# Bi-directional ribosome scanning controls the stringency of start codon selection

Yifei Gu[1], Yuanhui Mao [1], Longfei Jia[1], Leiming Dong[1] & Shu-Bing Qian [1✉]

The fidelity of start codon recognition by ribosomes is paramount during protein synthesis. The current knowledge of eukaryotic translation initiation implies unidirectional 5′→3′ migration of the pre-initiation complex (PIC) along the 5′ UTR. In probing translation initiation from ultra-short 5′ UTR, we report that an AUG triplet near the 5′ end can be selected via PIC backsliding. Bi-directional ribosome scanning is supported by competitive selection of closely spaced AUG codons and recognition of two initiation sites flanking an internal ribosome entry site. Transcriptome-wide PIC profiling reveals footprints with an oscillation pattern near the 5′ end and start codons. Depleting the RNA helicase eIF4A leads to reduced PIC oscillations and impaired selection of 5′ end start codons. Enhancing the ATPase activity of eIF4A promotes nonlinear PIC scanning and stimulates upstream translation initiation. The helicase-mediated PIC conformational switch may provide an operational mechanism that unifies ribosome recruitment, scanning, and start codon selection.

[1] Division of Nutritional Sciences, Cornell University, Ithaca, NY 14853, USA. ✉email: sq38@cornell.edu

Translation of eukaryotic mRNAs typically begins with the attachment of a 43S PIC to the capped 5′ end, which is facilitated by a heterotrimeric complex eIF4F comprised of eIF4E, eIF4A and eIF4G[1–3]. While our knowledge of translation initiation is steadily increasing, molecular details of many key steps remain to be hammered out. During the first step of translation, for instance, how mRNA is accommodated into the ribosome remains surprisingly obscure. Studies using in vitro reconstituted translation proposed a threading model[4], whereby the cap-binding protein eIF4E sits near the entry site and feeds the 5′ mRNA cap into the mRNA channel (Fig. 1a). This hypothesis was supported by the observation that 5′ UTR has no blind spots during scanning[4]. Although the threading model permits base-by-base inspection of mRNA from the 5′end, it does not explain the poor recognition of start codons near the 5′ end. The slot-in model, by contrast, directly places eIF4E near the exit site of the mRNA-binding cleft followed by lateral attachment of mRNA to the 40S (Fig. 1a). Perhaps the best evidence supporting the slotting mechanism comes from the internal ribosome entry site (IRES)-mediated cap-independent translation[5]. In addition, a 5′ UTR-tethered eIF4E enables ribosome recruitment independent of the cap structure[6]. For cap-dependent translation, however, the slot-in model is expected to leave a blind spot of ~12 nucleotides (nt) from the 5′ end. Another key difference between these two mutually exclusive pathways lies in the eIF4E kinetics

after mRNA loading. For the threading model to occur, eIF4E needs to be detached from the 5′ end cap of mRNA. The slot-in model, however, permits continuous eIF4E engagement at the 5′ end cap. While several lines of evidence supported cap-severed scanning[4,7,8], a recent study suggested cap-tethered ribosome scanning in human cells[9]. It remains unclear whether and how eIF4E dissociation from the cap occurs during mRNA accommodation.

Scanning commences once the PIC is loaded onto mRNA. It has been well-documented that stable hairpins in 5′ UTR inhibits translation initiation, especially when the secondary structure is close to the 5′ end[10]. The DEAD-box RNA helicase eIF4A is thought to prepare a single-stranded region near the 5′ end of mRNA, thereby facilitating PIC attachment[11]. However, recent studies suggest that yeast eIF4A enhances recruitment of mRNAs regardless of their structural complexity[12]. Notably, the ATPase activity of eIF4A is markedly stimulated in the presence of the PIC, suggesting a ribosome-dependent function of eIF4A beyond its role in RNA unwinding. Indeed, eIF4A appears to modulate the conformation of the PIC to promote mRNA accommodation in an ATP-dependent but helicase-independent manner[13]. It has been proposed that cycles of ATP hydrolysis of eIF4A lead to rounds of closed and open conformations of eIF4A, which correspond to alterations of closed and open states of the PIC. Despite the fact that eIF4A is the most abundant translation

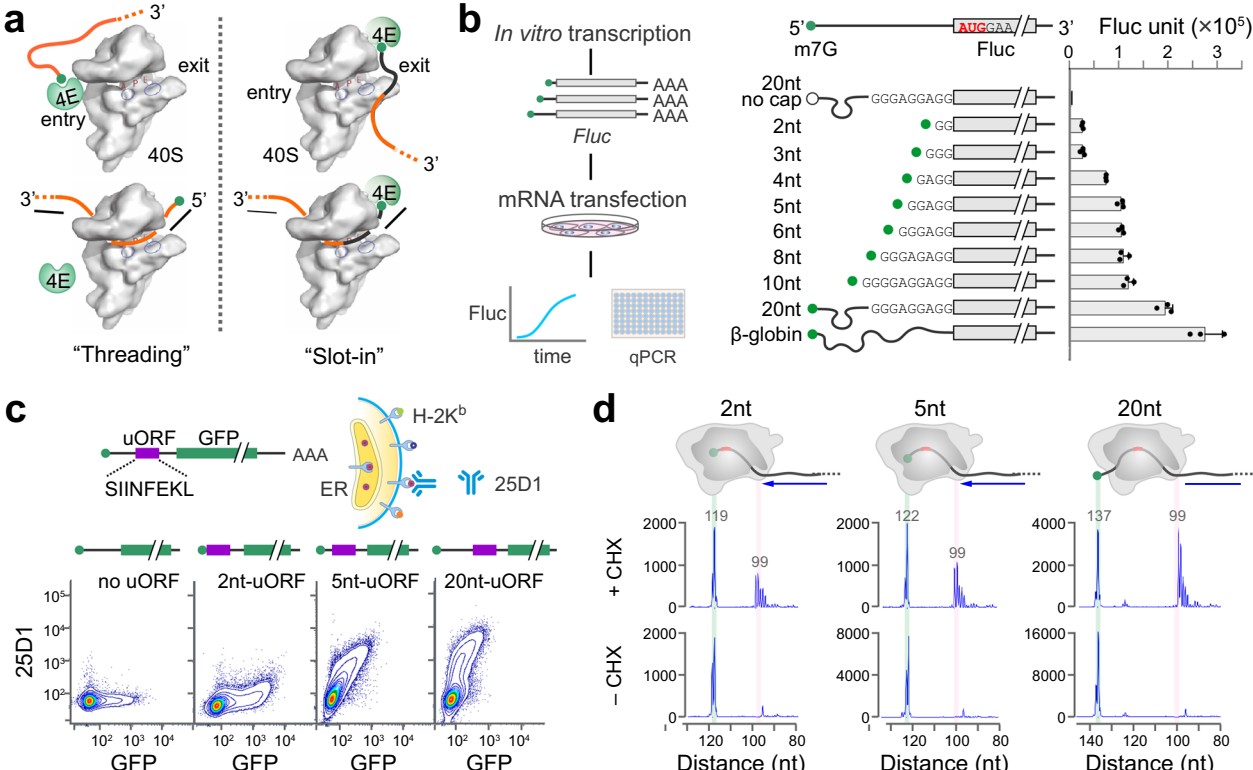

**Fig. 1 Selection of start codons near the 5′ end cap of mRNAs. a** Schematic of mRNA entering the 40S via the 'threading' mode (left) or the 'slot-in' mode (right). Only eIF4E is shown for simplicity. The black portion of mRNA indicates the blind spot. **b** Translation efficiency of Fluc mRNA reporters bearing ultra-short 5′UTR. The left panel shows the schematic of Real-time Luciferase Assay. In vitro synthesized mRNAs were transfected into MEF cells and Fluc activities were monitored by real-time luminometry. Fluc activities at 6 h were normalized to mRNA levels quantified by RT-qPCR. Uncapped mRNAs were included as negative control. Error bars: mean ± SEM; n = 3 biological replicates. **c** Translation efficiency of uORF mRNA reporters bearing ultra-short 5′ UTR. The top panel shows the schematic of uORF reporter assay coupled with FACS. The uORF reporter contains a sequence encoding SIINFEKL (purple) followed by GFP (green). Synthesized mRNA reporters were transfected into HEK293-K^b cells, followed by measurement of 25D1 and GFP using flow cytometry. The bottom panel shows the representative flow cytometry scatterplots of HEK293-K^b cells transfected with uORF reporters with a varied length of 5′ UTR. mRNAs with no uORF were included as negative control. **d** Start codon recognition by toe-printing assays. Fluc mRNA reporters with varied length of 5′ UTR were incubated in RRL, followed by reverse transcription using a probe downstream of the start codon. Expected positions corresponding to the full length and the ribosome leading edge are highlighted. Source data are provided as a Source Data file.

initiation factor[14], structural and biochemical data do not unanimously outline the position and orientation of eIF4A in the PIC complexes. As a result, the exact role of eIF4A in PIC loading and scanning remains incompletely understood.

The scanning process is arguably the least understood step of translation initiation. A widely accepted view indicates 5′→3′ directional movement of the PIC along 5′ UTR[15]. However, the underlying driving force has remained debatable and the determinants of the scanning directionality have been the subject of much speculation[16]. To complicate things further, noncanonical scanning modes like ribosome jumping or shunting have been documented[17–19]. Those cases of nonlinear scanning, albeit exceptional, serve as reminder against an over-simplified scanning model. Due to the highly dynamic nature of the process, a direct characterization of scanning has proven to be challenging. The recent development of translation complex profiling (TCP-seq) revealed heterogeneous footprint sizes for scanning PICs[7]. However, a snapshot of PIC footprints does not reveal the scanning directionality.

Proper recognition of mRNA start codons is coupled with the dynamic scanning process. It is commonly assumed that the first AUG codon that the scanning PIC encounters serves as the translation initiation site (TIS). This conventional wisdom has been challenged recently by many studies reporting alternative translation events initiated from multiple AUG codons as well as non-AUG triplets[20,21]. The stringency of TIS selection is influenced by cis-sequence elements and trans-acting factors, both of which affect the kinetics of PIC scanning. For instance, impediments to PIC scanning by stable hairpins enhance the recognition of suboptimal start codons that otherwise would be skipped[22,23]. On the other hand, even the most optimal start codon is not recognized with 100% efficiency. The subsequent leaky scanning implies the 5′ preference of start codon selection, which has long been attributed to the 3′-ward linear PIC migration. However, such 5′ polarity does not always hold true for closely spaced start codons[24], with downstream TIS negatively affecting the upstream start codon. The competitive nature of neighboring start codons suggests an intriguing possibility of bi-directional PIC scanning. Notably, backward scanning has been proposed during translation reinitiation[25,26], although the molecular dynamics of nonlinear migration remains poorly understood.

Here, coupling comprehensive mRNA reporters with genome-wide PIC and TIS profiling, we provide new lines of evidence supporting bi-directional ribosome scanning in mammalian cells. Unexpectedly, eIF4A appears to play a crucial role in driving nonlinear PIC scanning, thereby influencing the stringency of start codon selection. We propose that the ATPase-mediated conformational switch of ribosomes provides an operational mechanism that unifies lateral PIC recruitment, nonlinear scanning, and alternative start codon selection.

## Results

**Recognition of start codons near the 5′ end cap.** To address whether eukaryotic translation initiation follows threading or slot-in mechanisms for PIC attachment (Fig. 1a), we first attempted to determine the minimal length of 5′ UTR. Although in vitro studies have ruled out blind spots for start codon selection[4], whether mRNAs inside cells exhibit the same feature remains unclear. We constructed firefly luciferase (Fluc) reporters harboring varied length of 5′ UTR. To gain an accurate measurement of start codon selection, we (i) mutated the in-frame AUG codons of Fluc to minimize downstream initiation (Supplementary Fig. 1a); (ii) conducted transfection of mRNAs synthesized in vitro to eliminate transcript isoforms (Fig. 1b); (iii) employed a real-time luminometer to monitor Fluc levels above

the background (Supplementary Fig. 1b); (iv) performed RT-qPCR at end points to factor out variations of mRNA stability. Among reporters with increasing 5′ UTR length, a two-phase of translation efficiency is evident with the robust translation achieved when the 5′ UTR length reaches 20 nt or above (Fig. 1b, right panel). Intriguingly, Fluc mRNAs with ultra-short 5′ UTR (e.g., 2nt) are clearly translatable. The relatively low translation efficiency could be due to the suboptimal context missing the −3 purine.

For mRNA reporters with ultra-short 5′ UTR, the observed Fluc activities could be derived from downstream initiation using non-AUG codons. To rule out this possibility, we devised an independent reporter system bearing an upstream open reading frame (uORF) and a downstream GFP (Fig. 1c). The uORF-encoded tracer peptide (SIINFEKL) is efficiently presented on the cell surface by mouse major histocompatibility complex class I molecules H-2K$^b$ [27]. A monoclonal antibody 25D1 recognizes the K$^b$-SIINFEKL complex with exquisite sensitivity[28], thereby offering a direct measurement of the uORF product. Upon transfection into HEK293 cells stably expressing H-2K$^b$, both the tracer peptide and GFP can be quantified via flow cytometry. Like Fluc reporters, a 2-nt long 5′ UTR enables uORF translation as evidenced by the elevated 25D1 signals compared to the GFP control (Fig. 1c). We obtained similar results in cells transfected with plasmid-based reporters (Supplementary Fig. 1c). The translation of mRNAs with ultra-short 5′ UTR is strictly cap-dependent (Supplementary Fig. 1d), excluding the possibility of non-canonical ribosome entry. To confirm the exact positions of start codons selected by PIC, we conducted toe-printing assays using rabbit reticulum lysates (RRL) (Fig. 1d). Typical start codon-associated peaks appeared at the leading edge of initiating ribosomes immobilized by cycloheximide (CHX). We observed those peaks from the very first AUG triplet of all the mRNA templates regardless of the 5' UTR length. Therefore, start codons near the 5′ end can be reached by the PIC in vitro and inside cells, albeit with low efficiency.

**Tethering eIF4E prevents selection of start codons near the 5′ end cap.** The lack of an absolute blind spot in 5′ UTR for start codon selection is seemingly incompatible with the slot-in model. If the mRNA follows the threading model for start codon selection, tethering eIF4E to the 5′ end m7G cap would prevent the subsequent PIC attachment. We synthesized a Fluc reporter mRNA capped with m$^7$s$^6$G (Supplementary Fig. 2a), which could be crosslinked to its binding protein eIF4E under UV 360 nm (Fig. 2a)[29]. In the absence of crosslinking, m$^7$s$^6$G behaves like the normal m7G cap as evidenced by the comparable Fluc levels in RRL (Supplementary Fig. 2b). When purified eIF4E was cross-linked to the m$^7$s$^6$G cap, robust Fluc translation was readily observed for mRNAs bearing the β-globin 5′ UTR (Fig. 2b, top panel). The noticeable reduction could stem from non-specific effects of UV exposure. This result cannot be explained by the threading model because the constitutive presence of eIF4E at the 5′ end would prevent mRNA from entering into the narrow mRNA channel (Fig. 1a). Intriguingly, the non-dissociable eIF4E nearly abolished the translation of Fluc mRNAs with ultra-short 5′ UTR (Fig. 2c, top panel). To exclude possible side effects of UV crosslinking, we applied a chemical crosslinking methodology originally designed to identify eIF4E (Supplementary Fig. 2c)[30], and obtained similar results (Fig. 2b, c, bottom panel). The re-appearance of the blind spot when the 5′ cap is tethered with eIF4E suggests that mRNA is loaded into the PIC via slot-in.

**Competitive selection of closely spaced start codons.** Following the slot-in model, an immediate question is how the PIC loaded

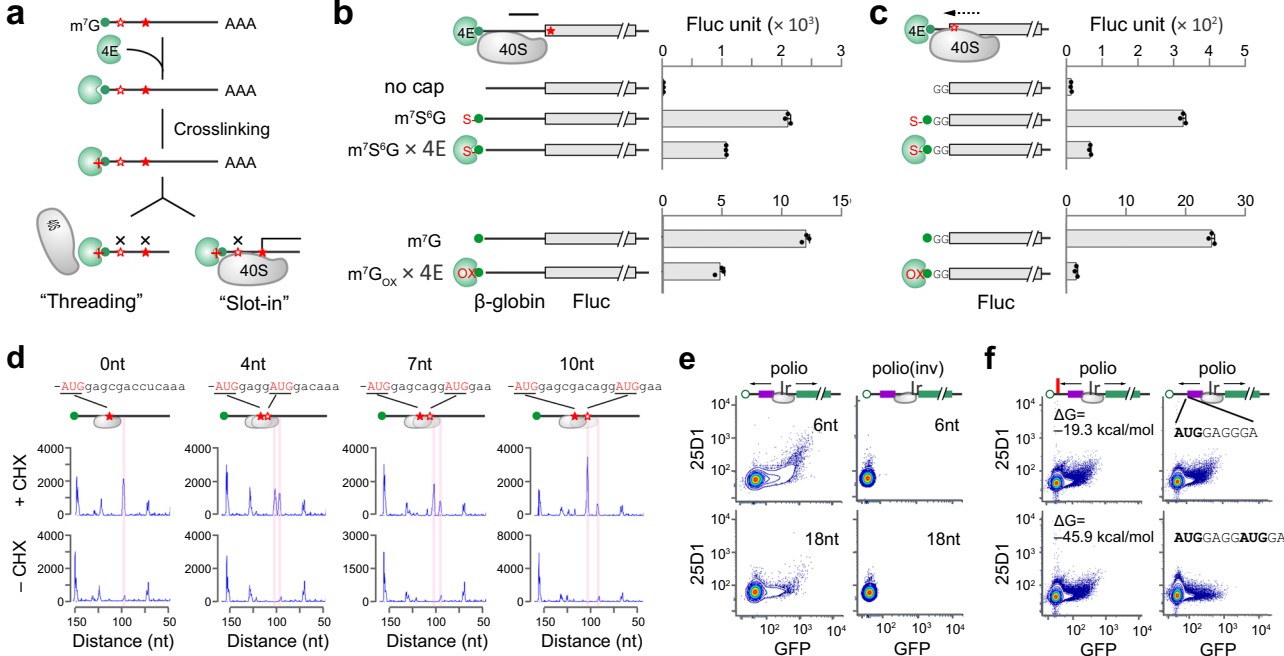

**Fig. 2 Bi-directional PIC scanning contributes to start codon selection. a** Schematic of cross-linking between m⁷G cap and eIF4E. Tethering eIF4E to m⁷G cap will prevent start codon selection in the 'threading' model, while permitting recognition of start codon downstream of the 'blind spot' in the 'slot-in' model. **b** Translational effects of cross-linking between m⁷G cap and eIF4E using Fluc mRNAs with β-globin 5′ UTR. The top panel used $m^7s^6G$ capped mRNAs followed by cross-linking under UV 360 nm. The bottom panel used redox reactions for cross-linking. In vitro translation was conducted in RRL and Fluc activities were monitored by luminometry. Error bars; mean ± SEM; $n = 3$ biological replicates. **c** Translational effects of cross-linking between m⁷G cap and eIF4E using Fluc mRNAs with ultra-short 5′UTR (2 nt). The top and bottom panels are similar to (**b**). Error bars; mean ± SEM; $n = 3$ biological replicates. **d** Selection of closely spaced AUG codons by toe-printing assays. Two AUG codons with identical flanking sequences but different length of the spacer were inserted into Fluc mRNAs with β-globin 5′UTR. Expected positions corresponding to the ribosome leading edge are highlighted. The results are representative of three biological replicates. **e** Selection of start codons flanking the IRES. The polio IRES was inserted between the uORF encoding SIINFEKL and GFP with 6 nt or 18 nt downstream of the uORF stop codon. The mRNA reporters were capped with non-functional ApppG. The flow cytometry scatterplots of HEK293-K^b cells transfected with mRNA reporters are representative of three biological replicates. **f** For the mRNA reporters as in (**e**), a stem loop with different stability was placed near the 5′ end (left panel) or an out-of-frame AUG was inserted downstream of the main AUG. The flow cytometry scatterplots of HEK293-K^b cells transfected with mRNA reporters are representative of three biological replicates. Source data are provided as a Source Data file.

downstream can select an upstream AUG triplet within the blind spot. One possibility is that the PIC relies on 3′→5′ movement to scan the 5′ end region. The backward excursion of ribosomes has been suggested from experiments using closely spaced AUG codons found in Turnip yellow mosaic virus (TYMV)[24]. We conducted similar experiments using toe-printing assays to quantify the relative usage of two AUG codons having identical flanking sequences (Fig. 2d). Indeed, placing an AUG codon 4 nt downstream reduced the selection of the first AUG. Importantly, with increasing spacer lengths, we observed a trend of ascending recognition of the first AUG and descending recognition of the second (Fig. 2d). The competitive relationship between neighboring AUG codons is incompatible with the linear scanning model. Only when the scanning ribosome undergoes back-and-forth oscillations, could the downstream AUG codon influence the upstream one. To substantiate this finding further, we placed three AUG codons into the blind spot with a single G in between. This design ensures that all AUG triplets are in different reading frames while retaining the same sequence context (Supplementary Fig. 2d). It is clear that AUG3 exhibited the most efficient initiation (>3 fold) in transfected cells. We also constructed mRNA reporters with the uORF driven by each AUG codons and obtained similar results (Supplementary Fig. 2e). Unlike the mRNA threading model that would follow the first-AUG rule in start codon selection, the more favorable recognition of the AUG3 supports reverse scanning of the PIC after slot-in.

To directly demonstrate that the PIC is capable of bi-directional scanning, we constructed uORF reporters by inserting an internal ribosome entry site (IRES) element between the tracer peptide SIINFEKL and GFP (Supplementary Fig. 2f). The poliovirus IRES was chosen because the class I IRES element follows the canonical initiation mechanism for PIC assembly and scanning[5]. In the absence of IRES, the non-functional cap analog ApppG supported neither uORF nor GFP translation in transfected HEK293-K^b cells (Supplementary Fig. 2g). Remarkably, the presence of the polio-IRES element after the uORF not only mediated downstream GFP translation, but also enabled uORF translation as evidenced by the elevated 25D1 signals (Fig. 2e). To rule out the possibility that the uORF translation was a result of forward scanning of the PIC looped from IRES, we reversed the orientation of the polio-IRES element (Supplementary Fig. 2f). In spite of the identical sequence and similar mRNA stability in transfected cells (Supplementary Fig. 2h), neither uORF nor GFP showed any translational activity (Fig. 2e). When the distance between uORF and IRES was increased from 6 nt to 18 nt, the translation potential of uORF was reduced. This result supports the backward migration of PIC recruited from the IRES.

**Transcriptome-wide PIC profiling supports bi-directional scanning.** A previous study developed translation complex profiling (TCP-seq) to track scanning ribosomes in budding yeast[7]. We reasoned that the pattern of PIC footprints near the 5′ end of

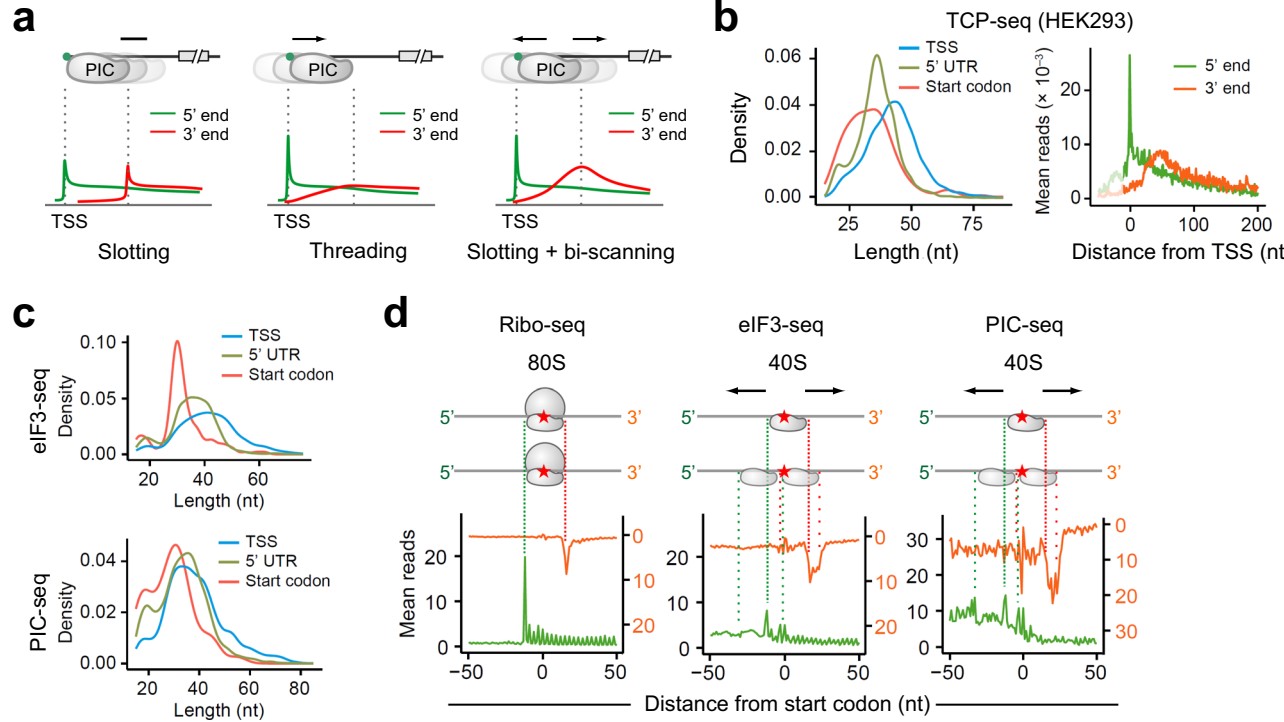

**Fig. 3 Genome-wide PIC profiling supports bi-directional scanning. a.** Schematic representation of threading, slotting, and slotting + bi-scanning models for PIC loading (top panel) and their corresponding read distribution of 5′ end (green) and 3′ end (red) of footprints. **b.** Reanalysis of TCP-seq data sets obtained from HEK293 cells (Wagner et al). The left panel shows the length distribution of reads mapped to TSS, 5′ UTR, or near start codons. Two-sided Wilcoxon test, all $P$ values <2.2 × 10$^{-16}$. The right panel shows the aggregation plots of 5′ end (green) and 3′ end (red) of reads on mRNAs aligned to TSS. **c.** MEF cells were used for eIF3-seq (top panels) and PIC-seq (bottom panels) followed by analysis of read length distribution. Two-sided Wilcoxon test, all $P$ values except for the start codon of PIC-seq <2.2 × 10$^{-16}$. For start codon vs TSS of PIC-seq, the $P = 6.9 × 10^{-8}$. **d.** Characterization of ribosome footprints from Ribo-seq, eIF3-seq, and PIC-seq data sets in MEF cells. Both the 5′ end (green) and 3′ end (red) of reads were mapped to transcripts aligned to annotated start codons (red star). Source data are provided as a Source Data file.

transcripts might help distinguish different modes of mRNA accommodation. For instance, the slot-in model is expected to leave a uniform width of PIC footprints from the 5′ end cap, resulting in the parallel distribution of 5′ and 3′ end of reads (Fig. 3a). The threading model, by contrast, gives rise to increasing lengths of PIC footprints from the transcription start site (TSS), forming a 5′ end peak only. Bi-directional scanning after slot-in, however, would form a 3′ end bump in addition to the 5′ end peak at TSS. This is because the presence of the cap-binding protein eIF4E delays PIC back-sliding until its dissociation from the 5′ end cap. By contrast, the limiting step of threading is expected to be near the TSS until the 5′ end cap emerges from the mRNA tunnel. In querying the TCP-seq data sets obtained from human cells[31], we noticed that the size of PIC footprints near the mRNA cap (or TSS) is surprisingly broader than that in 5′ UTR (Fig. 3b, left panel). The longer footprint size of the PIC near the TSS argues against the threading model. It is likely that the initiation complex loaded downstream of the 5′ end cap spans a broader mRNA region as expected from the presence of eIF4E. Mapping the 3′ end position of PIC footprints revealed a clear bump (Fig. 3b, right panel). The same feature holds true for the original TCP-seq conducted in budding yeast (Supplementary Fig. 3a). A recent study reported 40S profiling in zebra fish using a similar approach called ribosome complex profiling (RCP-seq)[32]. Despite the similar findings as TCP-seq, the authors concluded that the mRNA recruitment follows the threading model. We found that the same results could be interpreted as bi-directional PIC scanning after slot-in (Supplementary Fig. 3b). To substantiate this finding further, we explored selective TCP-seq associated with eIF3B[31]. The eIF3-associated PIC footprints near

the mRNA 5′ end not only showed broader sizes, but also displayed a typical bump for the 3′ end of reads (Supplementary Fig. 3c).

TCP-seq relies on sucrose gradient sedimentation to separate crosslinked 40S, which likely excludes initiation complexes with different sizes. To capture scanning PICs in a more comprehensive manner, we modified TCP-seq by enriching eIF3-associated ribosome complexes from mouse embryonic fibroblast (MEF) cells after formaldehyde crosslinking and RNase I digestion (Supplementary Fig. 3d). Compared to the standard Ribo-seq that shows a typical read length of ~29 nt, eIF3-seq uncovered a broader range of read length especially in 5′ UTR (Supplementary Fig. 3d). Like TCP-seq, eIF3 footprints near the TSS also showed a broader read length (Fig. 3c, top panel). We next examined the feature of eIF3 footprints near the annotated start codons, which could shed light on PIC dynamics during start codon selection. As expected, Ribo-seq showed a single 5′ and 3′ end peaks corresponding to the boundary of the 80S engaged at start codons (Fig. 3d, left panel). Intriguingly, eIF3 footprints showed several distinct peaks for 5′ and 3′ ends of reads (Fig. 3d, middle panel). These peaks were also evident in published TCP-seq data sets (Supplementary Fig. 3e). The eIF3 footprints flanking the start codon likely represent oscillating PICs averaging about 12 nt, although the heterogenous initiating ribosome complexes could also contribute to these footprints with varied 5′ and 3′ ends[9,31].

To distinguish eIF3:40S from eIF3:80S complexes, we subtracted 80 S footprints from the library of total RNA fragments to enrich PIC footprints (Supplementary Fig. 3d). Indeed, PIC-seq uncovered more 5′ UTR reads than eIF3-seq with broad read populations ranging from 19 nt to >60 nt. Once again, PIC-seq

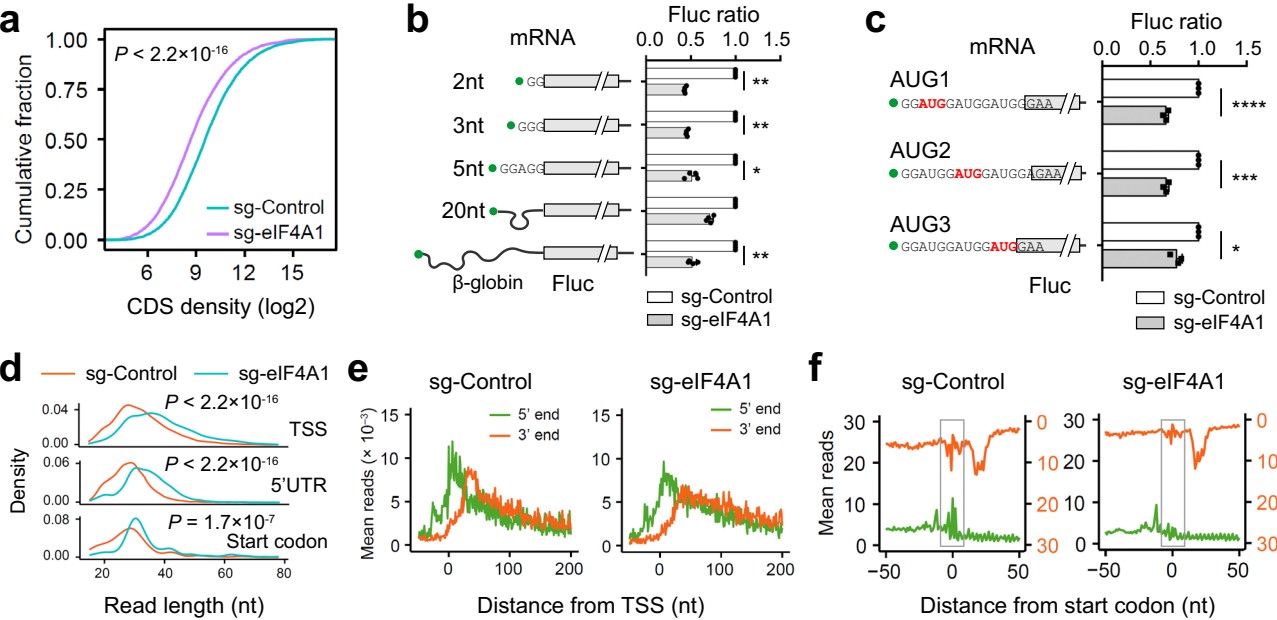

**Fig. 4 The RNA helicase eIF4A1 promotes free scanning of the PIC. a** Evaluation of global translation by Ribo-seq in MEF cells with or without eIF4A1 knockdown. Ribosome densities on individual transcripts were normalized to the number of mitochondrial reads and plotted as cumulative fractions. **b** Translation efficiency of Fluc mRNA reporters with ultra-short 5′ UTR in MEF cells with or without eIF4A1 knockdown. Fluc activities in the absence of eIF4A1 were normalized to the control. Error bars: mean ± SEM; n = 3 biological replicates. Two-tailed t-test, * p < 0.05; *** p < 0.001. **c**. Translation efficiency of Fluc mRNA reporters containing 3 AUG codons near the 5′ end in MEF cells with or without eIF4A1 knockdown. Fluc activities in the absence of eIF4A1 were normalized to the control. Error bars: mean ± SEM; n = 3 biological replicates. Two-tailed t-test, * p < 0.05; *** p < 0.001; **** p < 0.0001. **d** Comparison of read length distribution in cells with or without eIF4A1 knockdown. Reads mapped to TSS, 5′ UTR, or start codons are plotted separately. **e** Aggregation plots of 5′ end (green) and 3′ end (red) of reads on mRNAs aligned to TSS using eIF3-seq data sets obtained from cells with or without eIF4A1 knockdown. **f** Aggregation plots of 5′ end (green) and 3′ end (red) of reads on mRNAs aligned to start codons using eIF3-seq data sets obtained from cells with or without eIF4A1 knockdown. Highlighted squares indicate upstream footprints (3′ end of reads) and downstream footprints (5′ end of reads). Source data are provided as a Source Data file.

revealed a broader read length near the TSS than the start codon (Fig. 3c, bottom panel). When transcripts are aligned to the start codon, PIC footprints showed more prominent peaks downstream and upstream of the start codon (Fig. 3d). These results strongly support the oscillating behavior of PIC during start codon recognition, which potentially explains the competitive selection of neighboring start codons.

**eIF4A facilitates selection of start codons near the 5′ end cap**. It is unclear whether the bi-directional scanning of PIC is a spontaneous act or a guided event. The ATP-dependent helicase eIF4A has been widely believed to provide scanning directionality by resolving mRNA secondary structures in 5′ UTR[33]. We deleted *Eif4a1* from MEF cells using CRISPR/Cas9 gene editing and observed a marked disassembly of polysomes (Supplementary Fig. 4a and 4b). Consistently, both cell growth and global protein synthesis were reduced in the absence of eIF4A1 (Supplementary Fig. 4c and 4d). Ribo-seq also revealed global reduction of translation (Wilcoxon-test, $P < 2.2 \times 10^{-16}$) when the total amount of footprints was normalized to mitochondria (Fig. 4a), whose translation is independent of eIF4A. When individual mRNAs were considered, however, eIF4A1 depletion showed little mRNA specificity (Supplementary Fig. 4e). This finding agrees with the recent proposal that eIF4A promotes ribosome loading to nearly all mRNAs regardless of their structural complexity[12]. We next examined the role of eIF4A1 in the scanning process by measuring the translation of mRNA reporters with varied 5′ UTR length. Although silencing eIF4A1 globally reduced translation, the mRNA reporter bearing 20 nt 5′ UTR was relatively resistant to eIF4A1 knockdown (Fig. 4b and

Supplementary Fig. 4f). This is likely due to slot-in that directly locates the P-site of 40S near the start codon, thereby requiring minimal scanning. Notably, this feature was not seen in cells lacking eIF4A2 (Supplementary Fig. g and 4h). We further confirmed that, in mRNA reporters bearing 3 AUG codons near the 5′ end, AUG1 was more susceptible to eIF4A1 silencing than AUG3 (Fig. 4c and Supplementary Fig. 4i). These results collectively suggest that eIF4A1 facilitates PIC scanning in a bi-directional manner.

**eIF4A promotes free scanning of the PIC**. As a DEAD-box RNA-dependent ATPase, eIF4A1 has been shown to unwind RNA duplexes in a bi-directional manner[14,34]. Although the static view of PIC footprints is not suitable to infer the scanning directionality of PIC, the PIC conformation can be assessed by the length of footprints with an open conformation having shorter footprints[7]. Given the coordination between ATPase-dependent eIF4A and ribosome conformations[13], it is possible that the lack of eIF4A1 leads to more closed PIC conformation. To probe PIC conformation in the absence of eIF4A1, we conducted eIF3-seq and PIC-seq. Intriguingly, depletion of eIF4A1 resulted in longer reads in TSS and 5′ UTR (Fig. 4d). This was not due to sample variation because footprints mapped at the start codon have comparable read length. The longer PIC footprints in the absence of eIF4A1 is in line with the more closed PIC conformation. For PIC footprints near the TSS, the bump formed by the 3′ end of reads was reduced (Fig. 4e), presumably due to reduced PIC backsliding in the absence of eIF4A1. We next examined whether silencing eIF4A1 affects the PIC behavior in start codon recognition. Compared to the

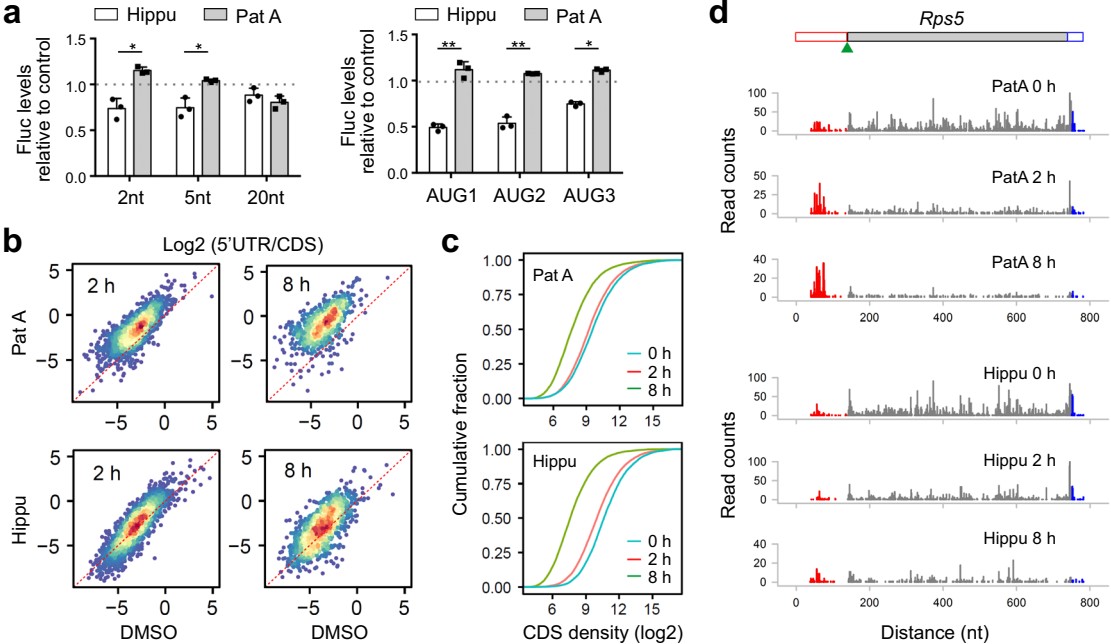

**Fig. 5 eIF4A promotes PIC scanning in an ATPase-dependent manner. a** Effects of Hippuristanol (Hippu) or Pateamine A (PatA) treatment on the translation efficiency of Fluc mRNA reporters with ultra-short 5′ UTR (top) or those containing three AUG codons near the 5′ end (bottom). Transfected MEF cells were treated with 200 nM Hippu or 50 nM PatA and Fluc activities were quantified by real-time luminometry. Error bars: mean ± SEM; n = 3 biological replicates. Two-tailed t-test, * p < 0.05; ** p < 0.01. **b** Effects of Hippu or PatA treatment on 5′ UTR ribosome occupancy revealed by Ribo-seq. Scatter plots show the correlation of read ratio (5′ UTR/CDS) between vehicle control and treatment (50 nM PatA: top; 200 nM Hippu: bottom) for 2 h (left) or 8 h (right). **c** Effects of Hippu or PatA treatment on CDS ribosome occupancy revealed by Ribo-seq. From the same data sets as (**b**), CDS ribosome occupancies on individual transcripts were normalized to the number of mitochondrial reads and plotted as cumulative fractions. **d** A representative gene (*Rps5*) shows differential 5′ UTR read density (red) relative to the CDS (dark gray) after treatment with eIF4A modulators. Source data are provided as a Source Data file.

control cells, eIF4A1 depletion nearly eliminated downstream reads (judged from 5′ end) and upstream reads (judged from 3′ end) relative to the start codon (Fig. 4f). These results strongly support the notion that eIF4A1 promotes free scanning of PIC by enabling an open conformation during start codon recognition. Without eIF4A1, the closed PIC conformation limits competitive selection of neighboring start codons.

**eIF4A promotes PIC oscillations in an ATPase-dependent manner.** We next seek the possibility of modulating the eIF4A activity using chemical compounds. The natural product hippuristanol (Hippu) potently inhibits the ATPase activity of eIF4A[35], whereas pateamine A (PatA) enhances its intrinsic enzymatic activities[36]. Pre-treatment with either Hippu or PatA readily disassembled polysomes (Supplementary Fig. 5a). Not surprisingly, both Hippu and PatA are commonly used as translation inhibitors despite their opposing effects toward the ATPase activity of eIF4A. We monitored the start codon selection from ultra-short 5′ UTR in MEFs after exposure to low concentrations of Hippu or PatA. Like eIF4A1 depletion (Fig. 4b), Hippu treatment repressed initiation of mRNAs with ultra-short 5′ UTR (Fig. 5a). Remarkably, pre-treatment with PatA promoted recognition of start codons near the 5′ end. The opposing effects of Hippu and PatA were no longer evident in the translation of messengers bearing 20-nt long 5′ UTR. We further examined mRNA reporters with 3 AUG codons near the 5′ end and found that the selection of those AUG codons from the blind spot was repressed by Hippu but not PatA (Fig. 5a, right panel). The finding that eIF4A enhances recognition of start codons within the blind spot argues against the hypothesis

that eIF4A prevents PIC from back-sliding. Rather, eIF4A promotes free scanning of the PIC in an ATP-dependent manner, presumably via an open conformation.

The average length of 5′ UTR in mammalian cells is ~200 nt[37]. It is thus unlikely that the bi-directional scanning of PIC is evolved to select start codons near the 5′ end. Since PIC oscillation occurs during the entire course of scanning, the back-and-forth excursions is expected to increase the dwell time of PIC in 5′ UTR and facilitate the recognition of upstream start sites (uTIS). We conducted Ribo-seq of MEFs exposed to eIF4A modulators for different times. To factor out variations of mRNA levels, we used the read density in the coding region (CDS) as internal control to compute 5′ UTR ribosome occupancy. Remarkably, PatA treatment for 2 h resulted in a global increase of 5′ UTR read density relative to the CDS (Fig. 5b, right panel). Notably, the overall CDS ribosome occupancy was reduced in the presence of PatA (Fig. 5c). This effect became more dramatic under prolonged PatA treatment (8 h). Typical examples are *Rps5* (Fig. 5d) and *Eef1a1* (Supplementary Fig. 5b). By contrast, Hippu treatment showed minimal changes of ribosome occupancy in 5′ UTR relative to CDS (Fig. 5b, bottom panel), despite its inhibitory effects on global translation (Fig. 5c). The opposing effect of Hippu and PatA on 5′ UTR read density was further confirmed when the 3′ UTR read density was used as internal control (Supplementary Fig. 5c). Since PatA enhances the ATPase activity of eIF4A, the marked accumulation of ribosome footprints in 5′ UTR is likely a consequence of increased recognition of uTIS sites. We noticed that, for some transcripts with few 5′ UTR reads, PatA treatment readily triggered the appearance of new footprints (Supplementary Fig. 5d), representing induced

**Fig. 6 Proposed models for eIF4A-mediated PIC conformational change that unifies ribosome recruitment, scanning, and start codon selection.** In the absence of eIF4A, the closed PIC conformation leads to reduced mRNA loading, delayed PIC scanning, and increased stringency of start codon selection. In the presence of eIF4A (orange triangle), multiple cycles of ATP hydrolysis lead to open conformation of PIC that facilitates slot-in mRNA loading, bi-directional PIC scanning, and selection of suboptimal start codons (empty red star).

alternative translation. A boost of eIF4A helicase activities thus offers a simple but effective means to uncover hidden uTIS sites across the transcriptome.

## Discussion

Eukaryotic translation begins with the accommodation of specific mRNAs by the 43 S PIC. The proposed mRNA threading mechanism permits base-by-base inspection by the PIC, enabling the selection of start codons near the 5′ end cap[4]. However, many transcripts with short 5′ UTRs have very low translation efficiency unless they bear a motif called translation initiator of short 5′ UTR (TISU)[38]. It is believed that TISU retains the PIC near the start codon after threading[39], although the slot-in mechanism is equally possible. With a 12 nt core sequence, TISU is nearly perfect for the ribosome A-site to land near the start codon after cap-dependent slot-in, permitting translation with minimal scanning. Another feature of the threading model is the mandatory eIF4E dissociation from the 5′ end cap. The potential role of eIF4E in TISU-mediated translation argues against the threading model[40]. We demonstrated that tethering eIF4E to the 5′ end cap does not abolish the translation of mRNAs with long 5′ UTR. A recent study using selective 40S foot-printing in human cells reported that eIF4E could remain attached on the PIC throughput the 5′UTR[9]. The slot-in model conforms to mRNA translation mediated by IRES[5], m6A residues[41,42], or the poly(A) tract in 5′ UTR[43]. The translation potential of circular RNAs (circRNAs) further indicate a more lateral loading of mRNA into the PIC[44]. For cap-dependent mRNA translation, however, the slotting PIC is expected to leave a blind spot on mRNA for start codon selection. In vitro studies have documented that a single nucleotide is a sufficient 5′UTR for translation[45]. Using in vivo mRNA reporters, we showed that a 5′ UTR of 2 nt enables mRNA translation albeit with low efficiency. It appears that neither the threading nor the slot-in model could fully explain those experimental observations.

It is possible that threading and slotting modes co-exist inside cells with the latter utilized solely for special types of mRNAs. However, the putative position of eIF4E in proximity to the E-site is inconsistent with the presumption of the threading model in which the cap-binding protein must be placed near the A-site. In addition, cross-linking assays revealed that the 5′ end of mRNA interacted with eIF3a, eIF3b, and eIF3d instead of small ribosomal proteins[46]. It is also worth noting that the PIC near the 5′ end shows a broad range of footprint sizes, which is more consistent with the laterally loaded PIC spanning a sizable mRNA region from the 5′ end cap. The major conflict between the threading and the slot-in model lies in the presence of blind spots for start codon selection. If the PIC undergoes a bi-directional motion after accommodation of mRNA, start codons near the 5′ end can be reached without threading. Supporting this notion, a blind spot emerged when the eIF4E was crosslinked with the 5′ end cap, which prevents PIC backsliding. We propose that PIC undergoes bi-directional migration after slotting, which resolves the

fundamental dilemma between slot-in mRNA accommodation and recognition of 5′ end start codons.

Another major open question in translation initiation concerns the scanning directionality. The linear correlation between the scanning time and the 5′ UTR length was demonstrated as the evidence supporting the 5′→3′ unidirectional movement of PIC[47]. For mRNAs with multiple start codons, the first AUG codon is favored by the scanning PIC also suggests the linear scanning mode[48]. Nevertheless, most of these conclusions considered a net directionality in the process but not the dynamics of the scanning. Despite the widely accepted scanning model, i.e., 5′→3′ linear and base-by-base migration, the nonlinear scanning mechanism has been suggested before. For instance, the IRES element from Halastavi arva virus (HalV) recruits PIC immediately downstream of the start codon, which is then picked by retrograde scanning[49]. The bidirectional scanning has also been documented for both post-terminating 80S ribosomes and recycled 40S during translation reinitiation[50,51]. For cap-dependent translation, the back-and-forth oscillations of scanning ribosomes was originally suggested from experiments using closely spaced AUG codons found in Turnip yellow mosaic virus (TYMV)[24]. We confirmed the competitive nature of neighboring start codons by demonstrating that downstream start codons negatively regulate upstream ones. Perhaps the most compelling evidence comes from mRNA reporters sandwiched by an IRES. Only when the PIC undergoes bi-directional scanning, could both upstream and downstream start codons be selected.

A more profound question is how the net 5′→3′ directionality is achieved on 5′ UTR if PIC undergoes back-and-forth movements. The DEAD-box RNA helicases have long been implicated in the scanning process. The yeast Ded1p catalyzes the forward scanning of PIC by residing at the mRNA entry channel and unwinding mRNA secondary structures[52]. The mammalian-specific DHX29 possibly uses a similar mechanism to assist forward scanning[53,54]. Not surprisingly, translation of mRNAs with complex structures is more susceptible to Ded1p or DHX29 deficiency. As the prototypical member of DEAD-box RNA helicase, eIF4A is the most abundant translation initiation factor[14]. Unlike DHX29 that is stably positioned at the leading edge of PIC, eIF4A is believed to recycle rapidly during the scanning process[11]. Notably, in vitro studies revealed that eIF4A can unwind RNA duplex in either direction at the same rate[55]. A previous study reported that eIF4A together with Rocaglamide A delays PIC scanning by clamping onto polypurine sequences in an ATP-independent manner[56]. In addition, eIF4A has been shown to modulate the PIC formations via ATP hydrolysis[13], suggesting that mRNA is not the only substrate of eIF4A. Indeed, lack of eIF4A leads to longer PIC footprints in 5′ UTR, a sign of closed PIC conformation. Consistently, in the absence of eIF4A, the closed PICs tend to stall in traversing 5′ UTR. As a result, global translation is severely inhibited. Given the crucial role of eIF4A in driving PIC free scanning, it is conceivable that the

nonlinear scanning consists of forward thrusts alternating with limited relaxation in the reverse direction (Fig. 6).

The scanning process is also crucial for the inspection of potential start codons on mRNAs. A growing body of evidence suggests that the stringency of start codon selection is tightly coupled with the scanning kinetics. In general, slow scanning enables a recognition of suboptimal start codons, whereas fast scanning increases the fidelity of start site selection. However, the PIC conformation is equally important, if not more, in controlling the stringency of start codon selection. Intriguingly, eIF4A-dependent changes in the conformation of eIF3j have been implied in the fidelity of start codon recognition[57]. By modulating the ATPase activity of eIF4A using small-molecule compounds, we demonstrate that recognition of cryptic uTIS sites can be induced or repressed. We anticipate that, by adjusting both ATPase and helicase activities of eIF4A, it is possible to fine-tune the stringency of start codon selection and control the translational output. As RNA helicases have been implicated in development and cancer[58], dissecting their distinct roles in the scanning process will accelerate our understanding of translational reprogramming in cellular homeostasis and pathophysiology.

## Methods

**Cell lines and reagents**. MEF, HEK293, and HEK293-K$^b$ cells were maintained in Dulbecco's Modified Eagle's Medium (DMEM) with 10% fetal bovine serum (FBS). Antibodies used in the immunoblotting are listed below: anti-eIF4A1 (Abcam ab31217), Anti-eIF4A2 (Life technologies, PA585142), anti-puromycin (Developmental Studies Hybridoma Bank PMY-2A4), anti-β-Actin (sigma A5441), Alexa Fluor 546 donkey anti-mouse secondary antibody (Invitrogen A10036) and Alexa Fluor 546 donkey anti-rabbit secondary antibody (Invitrogen A10040).

**Plasmid constructions**. The full-length firefly luciferase (Fluc) gene was cloned into pcDNA3.1 vector (Invitrogen) to generate the Fluc/pcDNA3.1 vector. To create the A88T mutant, site-directed mutagenesis was performed using the QuikChange II XL Site-Directed Mutagenesis Kit (Agilent Technologies). Primers are 5′- GCGTATCTCTTCAAAGCCTTATGCAGTTGCTCTCC -3′ and 5′- GGAGAGCAACGTGCATAAGGCTTTGAAGAGATACGC -3′. Mutation was confirmed by Sanger DNA sequencing. The uORF plasmid reporters were constructed by inserting DNA sequences containing 5′ UTR (2-nt, 4-nt, 10-nt or 20-nt) and the sequence encoding SIINFEKL into Sac I/BamH I sites of EGFP/pcDNA3.1 vector to generate SIINFEKL-EGFP/pcDNA3.1 vector. To construct the SIIN-FEKL-6nt-IRES-Fluc/pcDNA3.1 vector, the DNA sequence encoding SIINFEKL was cloned into Nhe I/ Pml I sites of Rluc-IRES-Fluc/ pcDNA3.1 vector. DNA sequence encoding EGFP was amplified from EGFP/pcDNA3.1 vector, and subcloned into BamHI/XohI sites of SIINFEKL-6nt-IRES-Fluc/pcDNA3.1 vector to generate SIINFEKL-6nt-IRES-EGFP/pcDNA3.1 vector. The SIINFEKL-18nt-IRES-EGFP/pcDNA3.1 vector was constructed by inserting 12nt + IRES sequence into Pml I/BamH I sites of SIINFEKL-6nt-IRES-EGFP/pcDNA3.1 vector. The Polio(inv) sequence was synthesized (Top Gene Technologies) and cloned into Pml I/BamH I sites of SIINFEKL-6nt-IRES-EGFP/pcDNA3.1 and SIINFEKL-18nt-IRES-EGFP/pcDNA3.1 vectors to generate SIINFEKL-6nt-IRES(inv)-EGFP/pcDNA3.1 and SIINFEKL-18nt-IRES(inv)-EGFP/pcDNA3.1 vectors, respectively. DNA sequences of all primers used in this study are listed in Supplementary Data 1.

**In vitro transcription**. To generate mRNAs suitable for transfection, 3 μg PCR products described above were utilized for in vitro transcription. Transcripts with the normal m$^7$G cap were generated using the mMESSAGE mMACHINE T7 Transcription Kit (Invitrogen 1344) and transcripts with the non-functional cap analog ApppG (NEB) were synthesized using MEGAscript T7 Transcription Kit (Invitrogen 1333), followed by polyadenylation using Poly(A) Tailing Kit (Invitrogen AM1350M). mRNA products were purified according to the manufacturer's instruction.

**Real-time luciferase assay**. Cells grown in 35 mm dishes were transfected with in vitro synthesized luciferase reporter mRNAs. Luciferase substrate D-luciferin (1 mM, Regis Tech) was added into the culture medium immediately after transfection. Luciferase activity was monitored and recorded using Kronos Dio Luminometer (Atto).

**Real-time quantitative PCR**. Total RNA was isolated by TRIzol reagent (Invitrogen) and used for reverse transcription assay via High Capacity cDNA Reverse

Transcription Kit (Invitrogen). Real-time PCR analysis was conducted using Power SYBR Green PCR Master Mix (Applied Biosystems) and carried on a LightCycler 480 Real-Time PCR System (Roche Applied Science). Primers for amplifying each target are listed in Supplementary Data 1.

**Flow cytometry**. Transfected HEK293-K$^b$ cells were washed with PBS and harvested by trypsin. Cells were then re-suspended in blocking buffer (1% bovine serum albumin (BSA) in PBS). Cells are aliquoted into a 96-well plate followed by 2000 rpm spinning for 2 min. After removal of blocking buffer, cells were washed one more time followed by staining with 25D1 Alexa 647 antibody (1:1000 in 75 uL solution per well). After incubation in the dark with gentle rocking at 4 °C for 30 min, cells were washed three times with 200 uL of the blocking buffer to remove unbound antibodies. Resuspend cells in 300 μl of blocking buffer followed by single cell filtering (Falcon). Cells were analyzed on a BD FACSAria Fusion flow cytometer (BD Biosciences). Cytometry data analysis is conducted using FlowJo.

**Toeprinting**. The ribosome binding reaction mixture was prepared on ice in a total volume of 10 μl containing 50% Nuclease-treated rabbit reticulocyte lysate (Promega), 20 pmol of primer (5′- 6-FAM-AATTGTTCCAGGAACCAG- 3′), 20 μM amino acid mixture, 0.4 U/ul RNaseOUT (Invitrogen) and 50 mM Tris-HCl pH 7.5. Reactions were treated with 0.5 mg/ml CHX followed by incubation at 37 °C for 5 min. After addition of 0.3 mg of reporter mRNAs, the reaction mixtures were incubated at 30 °C for 20 min to allow the translation machinery to assemble at the start codon. The reverse transcriptase reaction was conducted in a total volume of 20 μl containing the entire ribosome binding reaction, 1x Superscript III reverse transcriptase buffer, 5 mM DTT, 40 mM KCl, 3 mM MgCl$_2$, 50 mM Tris-HCl pH7.5, 0.5 mg/ml cycloheximide, 0.8 mM of dNTP, 1.5 U/μl RNaseOUT, 5 U/μl Superscript III reverse transcriptase. After incubation at 25 °C for 10 min, the reaction was terminated by nucleic acids extraction by phenol:chloroform and ethanol precipitation. The primer extension products were resuspended in 10 μl of Hi-Di formamide. 2 μl aliquot was run with 0.2 μl GeneScan 500 LIZ dye Size Standard (Fisher) on an ABI 3730xl instrument. Data is analyzed by Peak Scanner 2 software.

**Recombinant protein purification**. The *Eif4e* coding sequence was cloned into pGEX-6P-1 vector using the following primers:
eIF4E-F, 5′- GCGAATTCATGGCGACTGTCGAACCGGA-3′;
eIF4E-R, 5′-CCGCTCGAGTTAAACAACAAACCTATTTTTAG-3′.
The construct was transformed into the *E. coli* bacteria BL21. GST fusion protein was induced by 0.5 mM IPTG at 20 °C for 3–4 h. The cells were harvested and lysed in the lysis buffer (PBS supplemented with 0.5 mM PMSF, 1 mM DTT, protease inhibitor cocktail (Roche), 0.1% (v/v) Triton X-100) with 10 min sonication. Cell debris was removed by centrifuge at 12,000 rpm for 30 min. The supernatant was mixed with 2 ml equilibrated Pierce glutathione agarose followed by 4 °C incubation for 2-3 h. The resin was washed five times and the fusion protein was eluted in GST elution buffer (5 mM glutathione, 50 mM Tris-HCl pH 8.0).

**UV-crosslinking and chemical crosslinking**. mRNAs synthesized with the T7 RNA polymerase system (Invitrogen AM1333) were capped with GTP or 6-Thio-GTP using the capping enzyme system (CellScript C-SCCS1710), followed by polyadenylation with Poly(A) Tailing Kit (Invitrogen AM1350M). For UV-crosslinking, Purified GST-eIF4E protein was incubated with m$^7$S$^6$G capped mRNA in binding buffer (10 mM HEPES-KOH pH 7.6, 100 mM KCl, 0.5 mM MgCl$_2$, and protease inhibitors) at 4 °C under 360 nm UV exposure for 15 min. For chemical crosslinking, m$^7$G capped mRNA are incubated for 2–3 h on ice in 100 mM sodium acetate pH 5.3, 10 mM EDTA, 0.2 mM sodium periodate. The crosslinked mixture was neutralized by adding glycerol to 2% (v/v) final concentration. After 10 min incubation at room temperature, the crosslinked mRNAs were phenol extracted twice and ethanal precipitated. The RNA pellet was dissolved in Nuclease-free water. Purified GST-eIF4E protein was incubated with oxidized mRNA in binding buffer (10 mM HEPES-KOH pH7.6, 100 mM KCl, 0.5 mM MgCl$_2$, and protease inhibitors) at 4 °C for 15 min. NaBH$_4$ (20 mM) was added and incubation was pursued for 2–3 h on ice. UV or Chemical Crosslinked Mixtures were then incubated with 70% Nuclease-treated rabbit reticulocyte lysate (Promega), 20 mM amino acid mixture for 1.5 h at 30 °C. Firefly luciferase activities were measured using Luciferase Assay System (Promega E1500). Briefly, 20 μl of RRL mixture was transferred to each well of a 96-well plate. The 96-well plate was placed into the Plate-reading Luminometer with injector (BioTek). The injector added 100 μl of Luciferase Assay Reagent per well followed by the measurement of light intensity before the plate was advanced to the next well for a repeat of the inject-then-read process.

**Lentiviral sgRNAs**. LentiCRISPRv2 plasmids targeting eIF4AI were constructed using methods described previously[59,60]. Briefly, complementary oligonucleotides containing the specific sgRNA sequence and overhangs complementary to overhangs generated by BsmBI digestion of LentiCRISPRv2 were annealed to the BsmBI digested LentiCRISPRv2 plasmid to generate the functional transfer vector. Undigested LentiCRISPRv2 plasmid lacking a sgRNA sequence was used for

pseudovirus production as a control (Scramble). Lentiviral particles were packaged using Lenti-X 293 T cells (Clontech). Virus-containing supernatants were collected at 48 h after transfection and filtered to eliminate cells. MEF cells were infected by the lentivirus for 48 h followed by selection with 2 µg/ml puromycin. Selected cells were plated in 96-well plates by serial dilution at one cell per well. Single cells were then expanded and analyzed by PCR amplification of genomic DNA flanking the CRISPR-targeted region. sgRNA targeting sequence used in this study is listed in Supplementary Data 1.

**Puromycin labeling.** Cells at 80–90% confluence were treated with DMEM + 10% FBS 2 h before media was changed to DMEM + 10% FBS supplemented with 10 µg/ml puromycin for an additional 5 min, 10 min or 15 min. Cells were washed twice with ice-cold DPBS and lysed, followed by immunoblotting.

**Immunoblotting.** Cells were lysed on ice in TBS buffer (50 mM Tris pH7.5, 150 mM NaCl, 1 mM EDTA) containing 1% Triton X-100, 2 U/ml DNase and protease inhibitor cocktail tablet. The lysates were incubated on ice for 30 min, followed by heating for 10 min in SDS-PAGE sample buffer (50 mM Tris pH6.8, 100 mM dithiothreitol, 2% SDS, 0.1% bromophenol blue, 10% glycerol). Proteins were separated on SDS-PAGE and transferred to PVDF membranes (Fisher). Membranes were blocked in TBS containing 5% non-fat milk and 0.1 % Tween-20 for 1 h, followed by incubation with primary antibodies overnight at 4 °C. After incubation with horseradish peroxidase-coupled secondary antibodies at room temperature for 1 h, immunoblots were visualized using enhanced chemiluminescence (ECLPlus, GE Healthcare).

**eIF3-seq and PIC-seq.** A total of five 10 cm dishes with 90% confluence of MEF cells were used for either eIF3-seq or PIC-seq. After cell washing using ice-cold DPBS, cells were fixed in 10 ml/dish ice-cold formaldehyde solution (0.5% formaldehyde in DPBS) followed by 10 min incubation at 4 °C on a rocker. The cells were then washed with ice-cold DPBS and quenched in 10 ml/dish ice-cold buffer (50 mM Glycine, 50 mM Tris-HCl pH 7.5 in Nulease-free water) for 10 min at 4 °C on a rocker. After removal of the quench buffer, cells were washed with polysome buffer (10 mM HEPES-KOH pH 7.4, 100 mM KCl, and 5 mM MgCl₂), and collected in 400 µl lysis buffer (1% Triton-X-100 in polysome buffer) on ice. Whole cell lysates were cleared at 15,000 rpm for 10 min at 4 °C. The supernatant was transferred into a 1.5 ml tube followed by RNA digestion with RNase I (Ambion, 750 U per 100 A260 units). The mixture was incubated for 45 min at 4 °C. During the incubation, the sucrose solutions were prepared in polysome buffer. 10%–40% (w/v) sucrose density gradients were freshly prepared in a SW41 ultracentrifuge tube (Backman) using a Gradient Master (BioComp Instruments). Digested supernatant was loaded onto sucrose gradients followed by centrifugation for 3 h 30 min at 32,000 rpm 4 °C in a SW41 rotor. Separated samples were fractionated at 1.5 ml/min through an automated fractionation system (Isco) that continually monitors OD254 values. After separation, for eIF3-seq, 40 S and 80 S fractions were pool together and mixed thoroughly to get 600 µl total sample. 5 µg/mg lysate eIF3a antibody (Cell signaling 3411 S) and 0.5 U/ µl SUPERase•In (Invitrogen AM2696) were added to the IP samples, followed by incubation under gentle rotation at 4 °C for 3 h. Washed (3 times) Protein A/G beads were added into each IP sample and rock IP samples at 4 °C overnight. Beads were collected by spinning at 1000 rpm for 3 min at 4 °C and the supernatant was removed. Beads were washed 3 times with polysome buffer. After last washing, all supernatant was carefully removed and 400 µl of polysome buffer were added to resuspend the beads. For PIC-seq, 40 S and 80 S fractions were pool separately to get 400 µl each. For eIF3-seq, RNA was extracted from resuspended beads in polysome buffer. For PIC-seq, 40 S and 80 S were pooled together using methods previously described[61]. Briefly, samples were adjusted to 10 mM Tris-HCl pH 7.4, 10 mM glycine, 1% (w/v) SDS and 10 mM EDTA pH 8.0 and incubated at 65 °C for 5 min. Add one volume of acidic phenol/chloroform solution and vortex at maximum speed for 2 min. Place the tubes with mixtures into thermomixer and continue shaking at 1400 rpm for 20 min at 65 °C to reverse the cross-links. Centrifuge the samples at 14000 rpm for 5 min at room temperature. Carefully transfer the aqueous phases to new 1.5 ml tubes. Perform ethanol precipitation of RNA. Purified RNA was used for cDNA library construction and high-throughput sequencing described below.

**Ribo-seq.** For Ribo-seq, five 10 cm dishes of cells were harvested in 450 µl lysis buffer (1% Triton X-100 in polysome buffer) containing CHX (100 µg/ml), then centrifuged at 12,000 g 4 °C for 10 min. The supernatant was collected and subjected to sucrose gradient sedimentation. Sucrose solutions were prepared in polysome buffer. 15%- 45% (w/v) Sucrose density gradients were freshly prepared in a SW41 ultracentrifuge tube (Backman) using a Gradient Master (BioComp Instruments). Supernatant was loaded onto sucrose gradients followed by centrifugation for 2 h 30 min at 32,000 rpm 4 °C in a SW41 rotor. Separated samples were fractionated at 1.5 ml/min through an automated fractionation system (Isco) that continually monitors OD254 values. For both QTI-seq and Ribo-seq, ribosome fractions separated by sucrose gradient sedimentation were pooled and digested with E. coli RNase I (Ambion, 750 U per 100 A260 units) by incubation at 4 °C for 1 h. SUPERase•In (50 U per 100 U RNase I) was then added into the reaction mixture to stop the digestion. Total RNA was extracted using TRIzol LS reagent.

Purified RNA was used for cDNA library construction and high-throughput sequencing described below.

**cDNA library construction.** Fragmented RNAs were separated on a 15% polyacrylamide TBE-urea gel (Invitrogen) and visualized using SYBR Gold (Invitrogen). Selected regions of the gel corresponding to 20–70 nt (for PIC-seq and eIF3-seq) or 25-35 nt (for Ribo-seq and QTI-seq) were excised. The gel slices were disrupted by using centrifugation through the holes at the bottom of the tube. RNA fragments were dissolved by soaking overnight in 400 µl RNA gel elution buffer (300 mM NaOAc pH 5.5, 1 mM EDTA, 0.1 U/µl SUPERase•In). The gel debris was removed using a Spin-X column (Corning), followed by ethanol precipitation.

Purified RNA fragments were re-suspended in Nuclease-free water, then dephosphorylated for 1 h at 37 °C in a 15 µl reaction (1× T4 polynucleotide kinase buffer, 10 U SUPERase•In and 20 U T4 polynucleotide kinase). Dephosphorylated RNA fragments were precipitated using ethanol and re-suspended in Nuclease-free water. 0.15 µg linker (rApp/NNNNCTGTAGGCACCATCAAT/3ddC) then was added to the RNA fragments, heated at 70 °C for 90 s and then cooled to room temperature, followed by ligation for 3 h at at 22 °C in a 20 µl reaction (1× T4 Rnl2 reaction buffer, 10 U SUPERase•In, 15% PEG8000 and 20 U T4 RNA ligase 2 truncated). The reaction was heat inactivated at 80 °C for 10 min and the products were separated on a 10% polyacrylamide TBE-urea gel and selected regions in the gel corresponding to 45–95 nt (for PIC-seq and eIF3-seq) or 60-70 nt (for Ribo-seq and QTI-seq) were excised. RNA fragments were dissolved by soaking overnight in 400 µl RNA gel elution buffer. RNA fragments were Purified from RNA gel elution buffer as described earlier and re-suspended in Nuclease-free water.

For reverse transcription, the following oligos containing barcodes were used:
(Phos)
CTANNNAGATCGGAAGAGCGTCGTGTAGGGAAAGAGTGTAGATCTCGG
TGGTCGC(SpC18)CACTCA(SpC18)
TTCAGACGTGTGCTCTTCCGATCTATTGATGGTGCCTACAG
(Phos)
AGCNNNAGATCGGAAGAGCGTCGTGTAGGGAAAGAGTGTAGATCTCGG
TGGTCGC(SpC18)CACTCA(SpC18)
TTCAGACGTGTGCTCTTCCGATCTATTGATGGTGCCTACAG
(Phos)
ATTNNNAGATCGGAAGAGCGTCGTGTAGGGAAAGAGTGTAGATCTCGG
TGGTCGC(SpC18)CACTCA(SpC18)
TTCAGACGTGTGCTCTTCCGATCTATTGATGGTGCCTACAG
(Phos)
CCGNNNAGATCGGAAGAGCGTCGTGTAGGGAAAGAGTGTAGATCTCGG
TGGTCGC(SpC18)CACTCA(SpC18)
TTCAGACGTGTGCTCTTCCGATCTATTGATGGTGCCTACAGwhere Phos represents phosphorylation, NNN represents random sequence, SpC18 represents Hexa-ethyleneglycol spacer.

The linker ligated RNA sample was mixed with 0.5 mM dNTP and 2.5 mM synthesized primer and incubated at 75 °C for 3 min, followed by incubation on ice for at least 1 min. The reaction master mix was then added with 20 mM Tris (pH 8.4), 50 mM KCl, 5 mM MgCl₂, 10 mM DTT, 40 U RNaseOUT, and 200 U SuperScript III. Reverse transcription reaction was performed according to the manufacturer's instruction. Reverse transcription products were separated on a 10% polyacrylamide TBE-urea gel. Corresponding region was excised, which was expected to be approximately 200 nt. The first-strand cDNA products were recovered in DNA gel elution buffer (300 mM NaCl, 1 mM EDTA), then purified and re-suspended in Nuclease-free water as described earlier.

cDNA products were circularized in 20 µl of reaction containing 1×CircLigase buffer, 2.5 mM MnCl₂, 1 M Betaine, and 100 U CircLigase II (Epicentre). Circularization was performed at 60 °C for 1 h and the reaction was heat inactivated at 80 °C for 10 min.

**Deep sequencing.** Circular template was amplified by PCR using the Phusion high-fidelity (HF) enzyme (NEB) according to the manufacturer's instructions. The PCR forward primer: 5′-AATGATACGGCGACCACCGAGATCTACAC-3′ and reverse primer: 5′-CAAGCAGAAGACGGCATACGAGATGTGACTGGAGTTCA GACGTGTGCTCTTCCG -3′ were used to create products suitable for sequencing. PCR reaction contains 1× HF buffer, 0.2 mM dNTP, 0.5 µM oligonucleotide primers, and 0.25 U Phusion polymerase. PCR was carried out with an initial 30 s denaturation at 98 °C, followed by 12 cycles of 10 s denaturation at 98 °C, 20 s annealing at 65 °C, and 20 s extension at 72 °C. PCR products were separated on a nondenaturing 8% polyacrylamide TBE gel. Expected products around 180 bp were excised and recovered in DNA gel elution buffer, then purified and re-suspended in Nuclease-free water as described earlier[62].

After quantification by Agilent BioAnalyzer DNA 1000 assay, equal amounts of barcoded samples were pooled into one sample. Approximately 5 pM mixed DNA samples were used for cluster generation followed by sequencing by using sequencing primer 5′-CGACAGGTTCAGAGTTCTACAGTCCGACGATC-3′ (Illumina HiSeq).

**Alignment of sequencing reads.** The 3′ adapter CTGTAGGCACCATCAAT was trimmed by Cutadapt (5). The trimmed reads with length shorter than 15

nucleotides or longer than 35 nucleotides were discarded. The remaining reads were mapped to the mouse transcriptome using STAR with default parameters (6). Only the reads that were aligned to a unique position with mismatches lower than two nucleotides were kept. To construct mouse transcriptome, the annotation file and genome sequences downloaded from ENSEMBL database (GRCm38) were used. For each gene, the mRNA with longest CDS was selected. In the case of equal CDS length, the longest transcript was used.

**Ribosome density of transcript**. For each transcript, RPKM (reads per kilobase of CDS, per Million mapped reads) was used to estimate the ribosome density of transcript. To exclude the global effect of drug treatment, the count of total reads mapped to cytoplasmic transcriptome was first normalized by the count of reads that were aligned to mitochondrial genome. mRNAs with RPKM value < 1 were excluded.

**Aggregation plot of ribosome density**. To make the aggregation plot around start and stop codons, for each transcript, Ribo-seq reads at individual positions were normalized by the average reads of the transcript. Then, the normalized counts at the same position relative to start or stop codon were averaged. For 5′ end aggregation plot, positions of 5′ end of Ribo-seq reads were used. For 3′ end aggregation plot, positions of 3′ end of Ribo-seq reads were used.

**Reporting summary**. Further information on research design is available in the Nature Research Reporting Summary linked to this article.

## Data availability

The data that support this study are available from the corresponding author upon reasonable request. All sequencing data have been deposited in the Gene Expression Omnibus under accession number GSE176058. Source data are provided with this paper.

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

## Acknowledgements

We thank Dr. Jonathan W. Yewdell for providing 25D1 reagents and HEK293-K^b cells and Dr. Jerry Pelletier for the gift of Hippuristanol. We are grateful to Cornell University Life Sciences Core Laboratory Center for sequencing and FACS. This work was supported by US National Institutes of Health (R01GM1222814 and DP1GM142101) and HHMI Faculty Scholar (55108556) to S.-B.Q.

## Author contributions

S.-B.Q. conceived the project and designed the experiments. Y.G. performed the majority of experiments and Y.M. conducted the majority of data analysis. L.J. contributed to the uORF reporter assay. L.D. helped with library construction. S.-B.Q. wrote the manuscript. All authors discussed the results and edited the manuscript.

## Competing interests

The authors declare no competing interests.
