## [Peer Review File · Nature Communications]

Bi-directional ribosome scanning controls the stringency of start codon selectionReviewers' Comments:

Reviewer #1:

Remarks to the Author:

Gu et al. report on features of 5'UTR scanning by the pre-initiation complex during translation initiation on eukaryotic mRNAs. They find evidence for bi-directional scanning movement from experiments with reporter mRNAs carrying AUG start codons close to the cap, in part combined with covalently linking eIF4E to the cap, and mRNAs with multiple closely spaced AUG codons. Experiments involved reporter mRNA transfection into cells as well as toe-printing assays in vitro (figs 1&2). They then switch to transcriptome-wide ribosome footprinting, first re-analysing published TCP-seq data and then presenting their own variants of the approach called eIF3-seq and PIC-seq (figs 3-5). PIC footprint size, 5' and 3' end distribution in different 5'UTR/CDS locations indicate a model of bi-directional scanning after mRNA slot-in to the 40S subunit. Oscillating by the PIC is seen both, near the mRNA cap as well as around AUG start codons. Finally, using eIF4A1 deletion or manipulation of its activity by inhibitors, this oscillation is shown to be promoted by eIF4A in an ATP-dependent manner.

This manuscript is a mildly abbreviated version of a submission I recently reviewed for a different journal. Comparing the two versions, it is clear that, apart from the deletion of previous Fig 6, supplementary Figs S6,S7 and some associated text, they are essentially identical. Given that the deleted figures did not previously affect my assessment of the manuscript, I will simply restate my points made previously.

The authors provide a nice summary of relevant literature in their introduction and from this it is clear that this study is by no means the first attempt to conceptualise or experimentally assess the nature of scanning motion. It is however, in its use of different approaches and in-depth analysis, relatively unique and ultimately a break-through, even though I am not yet completely going along with the data interpretation as I understood it.

Approaches are diverse and valid; the quality of the data and its presentation is generally very high.

Statistics are well performed on the reporter data, but it is less clear to what extent the footprint sequencing data analyses have been checked for significance, e.g., difference of footprint size between regions, change in footprint 5' and 3' distribution upon manipulation of factors etc. I would suggest that this requires some good attention prior to publication.

Data with the reporter constructs appears generally very solid; although it is not clear if some of the flow cytometry and toe-printing panels are one-off data or were reproducible. Here, a systematic addition of reproducibility information to each panel legend, e.g., 'data is representative of X biological repeats', could solve the issue.

The picture is more mixed with the ribosome footprinting data. Part of the problem for me is the relative lack of statistical testing of differences that are then interpreted mechanistically. I'm also struggling to thoroughly assess some of the footprint meta-analyses against the known dimensions of the 40S subunit etc. The x-axis divisions in many panels, e.g., fig 3b,d or fig 4e,f, are just too coarse to really match fragment sizes to expected PIC footprints.

I am also quite worried about putting a lot of weight on footprint end metadata for positions exactly at the AUG codon as the sequenced fragments that begin or end there are not diverse in sequence and so the analyses could be vulnerable to library cloning and/or sequencing bias. I understand that fragments of this kind are reduced systematically in, e.g. eIF4A1 knockdown, but especially in the absence of statistical testing I remain worried here. Also, although the highlighted squares in fig 4f show concordant changes in 5' and 3' end abundance near position '0', shouldn't there be commensurate changes elsewhere in the orange and green lines as well?

The new sequencing method variants of eIF3-seq and PIC-seq are interesting but their explanation

and justification of their merit is a bit sparse (e.g., schematics are only in fig S3 rather than more prominently in the main figures). As presented, I am not too sure about some aspects of them as follows: Due to lack of sedimentation or other biophysical purification eIF3-seq will give a mix of 40S and 80S footprints as well as possibly some of eIF3 alone. PIC-seq is depleting all footprints found in regular 80S footprint sequencing but won't that not also deplete footprints of 40S where both 40 and 80S complexes are expected to be found, e.g., at AUG codons?

Minor improvement: Figure 1: b) How strong was the influence of stability measured this way? Does it match what might be gleaned from the functional half-life of the luc signal? Was the 'uncapped mRNA actually without cap or with an A-pppG cap as used elsewhere?

The introduction and discussion sections cite a range of relevant literature. One study that comes to mind and is of some relevance is De Gregorio et al. RNA 2001, 7: 106-113. It showed that even when tethering eIF4E internally to an mRNA in mammalian cells, productive translation ensues, excluding mandatory threading mediated by the factor. Another issue that could be attended better is that the PIC footprint 5'/3' end morphology found around start codons has been interpreted differently in previous TCP-seq papers, e.g., different intermediates of start codon recognition. How do the authors reconcile these with their new interpretations to do with bidirectional scanning? Or are these mutually exclusive?

The manuscript is structured and written very well in all its parts. There are some minor language issues that don't preclude understanding but could be ironed out to match grammar with the otherwise already high standard of writing.

Reviewer #2:

Remarks to the Author:

In this manuscript, Gu et al explore a very interesting topic, bidirectionality of pre-initiation complex (PIC) scanning. This is a largely unexplored topic and for that reason, I am supportive of the work. However, while the manuscript describes an overwhelming number of different experiments, most of them give somewhat ambiguous and weak support for the authors' interpretations. Some interpretations are very speculative and unconvincing.

In summary, the suggestion that initiation could take place at 5' UTRs as short as 2 nts (albeit inefficiently) should be taken seriously in light of the presented data, though I am not 100% convinced that it occurs in cells (in RRL it has been shown previously). I am less convinced by the data supporting scanning in the 3' to 5' direction and especially sceptical of the suggestion that this is how AUG codons are accessed at ultra short 5' UTRs, although I consider it plausible. The observations that PatA and Hippusinol have opposing effects on translation of ultra-short 5' UTRs is a very interesting one, but I am not at all convinced by the authors' explanations for this difference.

More specific comments:

1. Introduction.

The introduction is well written and provides a good account of currently debated aspects of initiation in eukaryotes, however, the information supporting backward scanning is limited to ref 23 which describes a violation of the leaky scanning rule when two AUG codons are close to each other. The main argument in support of backward scanning in ref. 23 is the downstream AUG influence on initiation at the upstream AUG. This has been observed in several other cases, e.g. in ref 22 recently (it is cited in the manuscript, but inappropriately), also in Dinesh-Kumar and Miller 1993 Plant Cell (10.1105/tpc.5.6.679), both groups of researchers suggested PIC stacking as an explanation of observed effect rather than backward scanning. This needs to be better represented in the introduction to put this work into a proper context along with Kozak et al 1995 PNAS

(10.1073/pnas.92.7.2662) who seemingly rejected the model that the authors are proposing here.

The other phenomenon relevant to backward scanning is its frequent occurrence during reinitiation. It was initially observed to occur on viral mRNAs (see section Reinitiation in Firth and Brierley (2012) *J Gen Virol* (10.1099/vir.0.042499-0), but subsequently was found to occur in cellular mRNAs as well, see Gould et al 2014 *RNA* (10.1261/rna.041574.113). It is well possible that bidirectional movements of ribosomes are also frequent in 5' UTRs where ribosomes often terminate at low efficient ORFs. I think this is worthy topic either for introduction or discussion

2. Minimal length of 5' UTRs supporting initiation. I do not understand how SINKFEL-GFP FACS experiments could be used as evidence that luciferase activity (Fig 1c) observed for short leader mRNAs does not come from low efficient non-AUG translation. In both cases, we see that very short leaders support very low initiation either of the luciferase or of SINKFEL synthesis. The first could occur due to the production of truncated luciferases from non-AUG initiation downstream, the second may occur due to low level transcription initiation upstream (as far as I understand, unlike luciferase reporters, GFP constructs are expressed from DNA and hence transcription artefacts cannot be excluded). The authors also map exact positions of initiating ribosomes, but in cell free system (RRL), where the functionality of ultra-short UTRs has been established (ref. 40). But RRL is somewhat a poor reference for cellular translation. Nonetheless, combined, the data obtained provide good support for the possibility of translation initiation at extra short 5' UTRs, albeit very inefficient.

3. Backward scanning. The experiments with SINKFEL-IRES-GFP are very provocative. However, the authors need to rule out the possibility that the effect that they observed is not due to changes in RNA levels. Poorly translated RNAs are often degraded more rapidly than highly translated. Thus, efficient translation from an active IRES may stabilize mRNA enabling detection of SINKFEL peptide even if its ORF translation efficiency does not change. A suggestion that SINKFEL uORF could be translated as a result of "forward scanning of the PIC looped from IRES" is an interesting one, but I do not understand how reversing polio IRES would rule it out. Probably the strongest support for the authors interpretation is that the increase of the spacer length between SINKFEL and GFP leads to SINKFEL reduction while GFP stays the same (right plots in fig 2e). However, I am not convinced that this is the case, looking at FACS plots, GFP is also reduced and for the plots on the right the variations for GFP activity between three replicates is very high to say with certainty that there is no difference in GFP activity. I would suggest repeating these experiments to increase statistical significance. But it is far more important to assess the RNA levels of these constructs. Given that an initiating ribosome complex at IRES may represent a roadblock to the scanning ribosomes it is also possible that initiation upstream may be stimulated by the PIC stacking as in ref 22 and in Dinesh-Kumar and Miller 1993 *Plant Cell* (10.1105/tpc.5.6.679). I am not sure how this could be ruled out, hence the authors at least should discuss this mechanism as an alternative to their chosen explanation.

4. Read length analysis. The authors consider slot-in and thread-in models as mutually exclusive, however, I don't see why we should exclude a possibility that both modes take place. This would explain the pattern observed in figure 3c without a need for backward scanning. Overall I find this analysis somewhat confusing and not convincing as asymmetric heterogeneity of footprint reads could occur due to the heterogeneity of the complexes, e.g. depending on the presence of a specific initiating factor a footprint may be shorter or longer either at the 5' or 3' side location. The authors interpretation of read lengths changes in eIF4A1 KO sounds too far reaching, there could be lots of different factors responsible for the observed changes.

5. eIF4A1 KO.

"Unexpectedly, messengers with ultra short 5'UTR were more sensitive to eIF4A1 depletion than those with long 5'UTR (Fig. 4b and Supplementary information, Fig. S4f)."

I don't see that at all in Fig. 4b. The statistical significance changes, among 2, 3 and 5 nt leaders, but this is due to technical variation, not due to real changes. Why reporters with longer UTRs are not

shown? Similarly for S4f, there are differences in activities of short UTRs for both KO and WT, but is this difference higher for shorter UTRs? Unclear if there is a difference between 2, 3 and 5 and longer ones are not shown.

6. Oscillations and the suggestion that backwards scanning increases translation at upstream starts. "Since PIC oscillation occurs during the entire course of scanning, the back-and-forth excursions is expected to increase the dwell time of PIC in 5'UTR and facilitate the recognition of upstream start sites (uTIS)."

I don't see how. Whether PIC moves unidirectionally (only forward) or bidirectionally (say two steps forward, one backward) the relative time that ribosome spend at downstream sites and upstream sites would be the same. Hence, such oscillations do not increase the chances of uTIS recognition. Nonetheless this

"Remarkably, PatA treatment for 2h resulted in a global increase of 5'UTR read density relative to the CDS (Fig. 5b, right panel)."

is a poor support for the above anyhow as increase of ribosome density in 5'UTRs is a general phenomenon observed in many stresses and could be explained in a number of ways. Increased amount of free ribosomes, reduced stringency of start selection due to subsequently limited number of available initiation factors are only some of many plausible explanations. Contrary to the authors descriptions the effect is seen for both PatA and Hippusinol, it is just more pronounced for PatA at earlier time points (Fig. 5b), hence I wonder if it relates more to the severity of the stress rather than the specificity of the compound.

Reviewer #3:

Remarks to the Author:

In the manuscript "Bidirectional ribosome scanning controls the stringency of start codon selection" the authors aimed to understand how ribosomes are recruited to mRNA and whether the process of PIC scanning is unidirectional or bi-directional. The study argues that ribosome are recruited by the slot-in mechanism and then that the PIC undergoes oscillations (backsliding) aided by the ATPase and helicase activities of eIF4A. The study is thorough and addresses an important unresolved problem in translation initiation. However, while we agree with the authors that the evidence for some backsliding is fairly convincing and that recruitment through slot-in is very likely to occur, we are not altogether convinced that the "threading" model can be entirely discarded. As also pointed out in the discussion section, the two models are not mutually exclusive. We also believe the manuscript would benefit from more thoroughly addressing previous results and evidence that has been presented in favor of this model and discuss how these should be interpreted under this new model.

Major issues:

- TISU elements are only very briefly mentioned in the discussion stating that these are "unlikely functional in the form of a linear thread". What is the evidence for this statement? The existence of TISUs is a significant argument in favor of threading and should therefore not be casually dismissed, but addressed through experiments to understand how the proposed backsliding model can explain the high efficiency of these cap-proximal initiators. Would you e.g. see initiation downstream of TISU elements as would be predicted by slot-in and backsliding? Would eIF4A1 KD have any effect? In general, the surrounding context of initiation codons seems to be somewhat ignored in the manuscript.

- Similarly, there is no mention of the studies of eIF1 which is the factor thought to inhibit translation initiation proximal to the cap under the "threading" model. A discussion of how to reconcile these prior

studies would be shed light on how to interpret the presented results.

- Fig 2b, shows a two-fold decrease in fluc activity on eIF4E crosslinking, which implies that translation is substantially inhibited by the tethering of the cap-protein (eIF4E). This "lost translation" could very well be occurring via the threading mechanism and it is therefore a too strong statement to claim that "This result cannot be explained by the "threading" model ...". It is only correct that it cannot be explained by "threading" being the ONLY mode of initiation.

- The same argument can be raised for Fig 2C where an alternate interpretation can be that the decrease in fluc activity for mRNAs with ultra-short 5'UTR upon eIF4E cross-linking is due to the mRNA not being able to enter the ribosomal mRNA channel through "threading".

- It is unclear why the "bump" in 3' ends is expected by slot-in followed bi-directional scanning. Is bidirectionally only occurring right after slot-in? If bi-directional scanning occurs at the same probability throughout the 5'UTR the 3' distribution should not necessarily differ close to the cap, it should rather increase up to the site of slot-in and from there be relatively uniform throughout the 5'UTR. It is also not immediately obvious that other models could not produce this pattern. A similar bump could result without back-sliding if slot-in is not precise down to a single base, but the 5' end remains protected, or under the threading model if the speed is not uniform and/or if there is a detectable loss of scanning complexes across the UTR.

- While the manuscript makes a point of the longer reads at the TSS it should also be noted that both TCP-seq and RCP-seq revealed a significant number of shorter fragments (which could probably be even more abundant if not size-selected and lost due to mapping issues with very short reads). While this is consistent with both backsliding and threading it would under the backsliding model suggest that a substantial(?) number of ribosomes slide entirely off the mRNA. At the very least this possibility should be addressed and discussed.

Minor issues:

Page 3, line 72: scanning PIC encounters "servers" should be serves

Page 3, line 82: negatively affects should be negatively affecting

Page 10, line 274: Fig.5a, left panel should be Fig.5a, right panel

Page 11, line 303: celled should be called

Supplementary fig 4h and 4i are not referred to in the main text.

Reviewer #1:

The authors provide a nice summary of relevant literature in their introduction and from this it is clear that this study is by no means the first attempt to conceptualise or experimentally assess the nature of scanning motion. It is however, in its use of different approaches and in-depth analysis, relatively unique and ultimately a break-through, even though I am not yet completely going along with the data interpretation as I understood it.

Approaches are diverse and valid; the quality of the data and its presentation is generally very high.

We thank the Reviewer for concisely summarizing our work. We are pleased to receive the Reviewer's positive comments on our work and the support on the concept of non-linear scanning mode of ribosomes during translation initiation.

1. Statistics are well performed on the reporter data, but it is less clear to what extent the footprint sequencing data analyses have been checked for significance, e.g., difference of footprint size between regions, change in footprint 5' and 3' distribution upon manipulation of factors etc. I would suggest that this requires some good attention prior to publication.

We appreciate the Reviewer's effort to improve our manuscript. We used multiple data sets for the ribosome footprint analysis. In the case of footprint size comparison, we first re-analyzed published data sets obtained from *S. cerevisiae* (TCP-seq), zebrafish (RCP-seq), and HEK293 (sel-TCP-seq). As presented in Supplementary Fig. 3, these results showed consistent pattern. Notably, these data sets were analyzed by combining replicates to increase the read depth. Similarly, for our own data sets (i.e., Ribo-seq, eIF3-seq, and PIC-seq), we also grouped all the replicates to increase the read depth. We believe that a higher read depth would convey more confidence than single replicates in drawing our conclusion.

To address the Reviewer's concern regarding the statistical analysis of footprints, we have conducted Wilcoxon test for Fig. 3b, Fig. 3c, Fig. 4d, Supplementary Fig. 3a-3c. We have added statistical values in the corresponding figure legends of the revised manuscript.

2. Data with the reporter constructs appears generally very solid; although it is not clear if some of the flow cytometry and toe-printing panels are one-off data or were reproducible. Here, a systematic addition of reproducibility information to each panel legend, e.g., 'data is representative of X biological repeats', could solve the issue.

We totally agree with the Reviewer regarding the importance of biological repeats in flow cytometry and toe-printing analysis. For flow cytometry, we routinely conducted at least three times. Together with the flow scatter, we presented a bar graph showing the mean fluorescence intensity of all the replicates (for instance, Fig. S2e). In terms of toe-printing results, we have followed the Reviewer's suggestion by including a statement in the figure legend.

3. The picture is more mixed with the ribosome footprinting data. Part of the problem for me is the relative lack of statistical testing of differences that are then interpreted mechanistically. I'm also struggling to thoroughly assess some of the footprint meta-analyses against the known dimensions of the 40S subunit etc. The x-axis divisions in many panels, e.g., fig 3b,d or fig 4e,f, are just too coarse to really match fragment sizes to expected PIC footprints.

We apologize for the lack of clarity in the figure presentation of ribosome footprints. As mentioned above, we combined replicates to increase the read depth. Unlike TCP-seq and RCP-seq that rely on sucrose gradient sedimentation to separate crosslinked 40S, PIC-seq used immunoprecipitation to capture initiation complexes. To address the Reviewer's concern, we plotted both sel-TCP-seq and PIC-seq together since both data sets were obtained from HEK293 cells. It is clear that the overall pattern is very much consistent although TCP-seq showed broader TSS footprints (**Figure 2** of this letter). In addition to the added statistical values, we have clarified this point in the main text of the revised manuscript.

Figure 2. A direct comparison of footprint size between PIC-seq (red) and TCP-seq (blue). Overall reads are stratified based on regions.

4. I am also quite worried about putting a lot of weight on footprint end metadata for positions exactly at the AUG codon as the sequenced fragments that begin or end there are not diverse in sequence and so the analyses could be vulnerable to library cloning and/or sequencing bias. I understand that fragments of this kind are reduced systematically in, e.g. eIF4A1 knockdown, but especially in the absence of statistical testing I remain worried here. Also, although the highlighted squares in fig 4f show concordant changes in 5' and 3' end abundance near position '0', shouldn't there be commensurate changes elsewhere in the orange and green lines as well?

The Reviewer raised an important point in terms of how to interpret footprints around the start codon. We agree with the Reviewer that some of the start codon-associated reads might be prone to sequencing bias. However, when the AUG is positioned at the P-site, both 5' and 3' ends have varied nucleotides. In Fig. 4f, we showed on purpose both 5' and 3' ends of all the mappable footprints to avoid possible artifacts.

As pointed out by the Reviewer, in cells lacking eIF4A1, the highlighted region showed an evident reduction of read density without concordant changes in other regions. This is an important observation because a uniform decrease would suggest a non-specific global effect. Since eIF4A controls the dynamics of initiating ribosomes, a more specific effect is expected to occur around the start codon. In Fig. 4f, the other region serves as an internal control for the start codon region. Notably, the shift of 5' peaks is consistent with the increased footprint size in the absence of eIF4A1 as shown

in Fig. 4d. In the revised manuscript, we have rephrased several sentences to ensure clarification.

5. The new sequencing method variants of eIF3-seq and PIC-seq are interesting but their explanation and justification of their merit is a bit sparse (e.g., schematics are only in fig S3 rather than more prominently in the main figures). As presented, I am not too sure about some aspects of them as follows: Due to lack of sedimentation or other biophysical purification eIF3-seq will give a mix of 40S and 80S footprints as well as possibly some of eIF3 alone. PIC-seq is depleting all footprints found in regular 80S footprint sequencing but won't that not also deplete footprints of 40S where both 40 and 80S complexes are expected to be found, e.g., at AUG codons?

We apologize for the lack of details in describing the PIC-seq procedure partly due to the space limitation. Notably, we did sedimentation on sucrose gradient before eIF3 immunoprecipitation of ribosome fractions. Therefore, it is unlikely that eIF3-seq contains a mix of 40S and 80S footprints as well as eIF3 alone.

Like TCP-seq, PIC-seq is designed to enrich 40S by 80S footprint depletion. This is particularly important for start codon footprints because, as pointed out by the Reviewer, both 40S and 80S complexes could be positioned at the same start codon. 80S footprint depletion greatly helps enrich the 40S footprints, which is a critical step of PIC-seq. As shown in Fig. 3d, PIC-seq generally has higher read density than eIF3-seq for 5'UTR region including the start codon. In the revised manuscript, we have further clarified this point by describing the procedure in more detail.

Notably, since other Reviewers raised similar confusions in the interpretation of PIC-seq results, we have now added a new paragraph in the main text to include other possible explanation.

Minor improvement:

6. Figure 1: b) How strong was the influence of stability measured this way? Does it match what might be gleaned from the functional half-life of the luc signal? Was the 'uncapped mRNA actually without cap or with an A-pppG cap as used elsewhere?

We appreciate the Reviewer's obvious care in reviewing our manuscript. We measured the Fluc mRNA levels in all transfected cells (6 hr post-transfection) and the difference is negligible. In Fig. 1b of the manuscript, the bar graph shows the Fluc activities already normalized with mRNA abundance. Additionally, the half-life of Fluc signals is not expected to affect the steady-state levels because we only changed the 5'UTR but not the Fluc coding region. It is worth noting that we conducted real-time luminometry to monitor Fluc levels that accumulate along with the transfection time.

With respect to the "uncapped" mRNAs in Fig. 1b, they indeed have no cap. To address the Reviewer's concern, we compared mRNAs without cap or with a non-functional cap analog ApppG. Although the no-cap control showed basal Fluc levels, ultra-short 5'UTR reporters capped with ApppG exhibited discernable Fluc activities (**Figure 3** of this letter). However, these levels are negligible in comparison to the functional m⁷G cap. Therefore, we are confident that translation of mRNA reporters with ultra-short 5'UTR is cap dependent.

Figure 3. Real-time luminometry of MEF cells transfected with different Fluc mRNA reporters. The right panel shows the amplified region of the left panel.

7. The introduction and discussion sections cite a range of relevant literature. One study that comes to mind and is of some relevance is De Gregorio *et al.* *RNA* 2001, 7: 106-113. It showed that even when tethering eIF4E internally to an mRNA in mammalian cells, productive translation ensues, excluding mandatory threading mediated by the factor. Another issue that could be attended better is that the PIC footprint 5'/3' end morphology found around start codons has been interpreted differently in previous TCP-seq papers, e.g., different intermediates of start codon recognition. How do the authors reconcile these with their new interpretations to do with bidirectional scanning? Or are these mutually exclusive?

We thank the Reviewer for altering us to this important reference, which is interesting and highly relevant. Further supporting the “slot-in” mode of mRNA loading, we are glad to have this opportunity to cite this work in the revised manuscript,

With regard to reconciling the PIC footprint around start codons with TCP-seq, the Reviewer’s comment is well-taken. The different intermediates of the initiation complex uncovered TCP-seq are not mutually exclusive with bidirectional scanning. Due to the read depth of TCP-seq, footprints can be stratified by fixing either 5’ or 3’ end, thereby revealing distinct intermediates. However, the varied 5’ and 3’ ends of footprints are also compatible with the bi-directional mode of ribosome scanning. We believe that it is more informative by showing both 5’ and 3’ end positions within the same plot, as exemplified in Fig. 3d. In the revised manuscript, we added relevant discussions in the main text to reconcile these results.

Reviewer #2:

In this manuscript, Gu et al explore a very interesting topic, bidirectionality of pre-initiation complex (PIC) scanning. This is a largely unexplored topic and for that reason, I am supportive of the work. However, while the manuscript describes an overwhelming number of different experiments, most of them give somewhat ambiguous and weak support for the authors' interpretations. Some interpretations are very speculative and unconvincing.

In summary, the suggestion that initiation could take place at 5' UTRs as short as 2 nts (albeit inefficiently) should be taken seriously in light of the presented data, though I am not 100% convinced that it occurs in cells (in RRL it has been shown previously). I am less convinced by the data supporting scanning in the 3' to 5' direction and especially sceptical of the suggestion that this is how AUG codons are accessed at ultra short 5' UTRs, although I consider it plausible. The observations that PatA and Hippusinol have opposing effects on translation of ultra-short 5' UTRs is a very interesting one, but I am not at all convinced by the authors' explanations for this difference.

We appreciate the Reviewer's supportive comments on our manuscript. We are fully aware that challenging the existing dogma is not an easy task and often requires rigorous experimental support. To this end, we have conducted several independent experiments to strengthen the central hypothesis. The Reviewer's comments are very insightful, and we have addressed them point-by-point below.

Specific comments**1. Introduction.**

The introduction is well written and provides a good account of currently debated aspects of initiation in eukaryotes, however, the information supporting backward scanning is limited to ref 23 which describes a violation of the leaky scanning rule when two AUG codons are close to each other. The main argument in support of backward scanning in ref. 23 is the downstream AUG influence on initiation at the upstream AUG. This has been observed in several other cases, e.g. in ref 22 recently (it is cited in the manuscript, but inappropriately), also in Dinesh-Kumar and Miller 1993 Plant Cell (10.1105/tpc.5.6.679), both groups of researchers suggested PIC stacking as an explanation of observed effect rather than backward scanning. This needs to be better represented in the introduction to put this work into a proper context along with Kozak et al 1995 PNAS (10.1073/pnas.92.7.2662) who seemingly rejected the model that the authors are proposing here.

The other phenomenon relevant to backward scanning is its frequent occurrence during reinitiation. It was initially observed to occur on viral mRNAs (see section Reinitiation in Firth and Brierley (2012) J Gen Virol (10.1099/vir.0.042499-0), but subsequently was found to occur in cellular mRNAs as well, see Gould et al 2014 RNA (10.1261/rna.041574.113). It is well possible that bidirectional movements of ribosomes are also frequent in 5' UTRs where ribosomes often terminate at low efficient ORFs. I think this is worthy topic either for introduction or discussion

We appreciate the Reviewer's obvious care in reviewing our manuscript. Those papers mentioned by the Reviewer are extremely informative and we are glad to have the opportunity to include these highly relevant literatures in both introduction and discussion. It is worth mentioning that the reporters used in our studies are distinct from prior studies in Dinesh-Kumar and Miller 1993 *Plant Cell* (10.1105/tpc.5.6.679). In our study, the two AUG codons were positioned close to each other to exclude the PIC stacking possibility, as reported by others. Regardless, in the revised manuscript, we have included prior studies in the main text and cited all the references mentioned by the Reviewer.

2. Minimal length of 5' UTRs supporting initiation. I do not understand how SINFKEL-GFP FACS experiments could be used as evidence that luciferase activity (Fig 1c) observed for short leader mRNAs does not come from low efficient non-AUG translation. In both cases, we see that very short leaders support very low initiation either of the luciferase or of SINFKEL synthesis. The first could occur due to the production of truncated luciferases from non-AUG initiation downstream, the second may occur due to low level transcription initiation upstream (as far as I understand, unlike luciferase reporters, GFP constructs are expressed from DNA and hence transcription artefacts cannot be excluded). The authors also map exact positions of initiating ribosomes, but in cell free system (RRL), where the functionality of ultra-short UTRs has been established (ref. 40). But RRL is somewhat a poor reference for cellular translation. Nonetheless, combined, the data obtained provide good support for the possibility of translation initiation at extra short 5' UTRs, albeit very inefficient.

We would like to clarify that we only used mRNA reporters for both luciferase and SIINFEKL measurement. As pointed out by the Reviewer, the transcription-associated variation does not give rise to definitive TSS of mRNAs. Additionally, we used mRNA reporters to avoid isoforms during transcription. We are also aware of the non-AUG initiation possibility that could contribute to the truncated luciferase. However, this cannot be the case for SIINFEKL because downstream initiation would destroy the peptide. In our previous study (Jia *et al.* *Nat Struct Mol Biol* 2020), we used the same SIINFEKL reporter to dissect start codon selection.

With regard to the cell-free system like RRL, we totally agree with the Reviewer that the *in vitro* system does not represent the cellular system. We were merely seeking an independent approach to validating the translatability from ultra-short 5'UTR. We appreciate the Reviewer's agreement that these *in vitro* and *in vivo* results collectively support the notion that mRNAs with extra short 5'UTR are still translatable.

3. Backward scanning. The experiments with SINFKEL-IRES-GFP are very provocative. However, the authors need to rule out the possibility that the effect that they observed is not due to changes in RNA levels. Poorly translated RNAs are often degraded more rapidly than highly translated. Thus, efficient translation from an active IRES may stabilize mRNA enabling detection of SINKFEL peptide even if its ORF translation efficiency does not change. A suggestion that SINKEL uORF could be translated as a result of "forward scanning of the PIC looped from IRES" is an interesting one, but I do not understand how reversing polio IRES would rule it out. Probably the strongest support for the authors interpretation is that the increase of the spacer length between

SINKFEL and *GFP* leads to *SINKFEL* reduction while *GFP* stays the same (right plots in fig 2e). However, I am not convinced that this is the case, looking at FACS plots, *GFP* is also reduced and for the plots on the right the variations for *GFP* activity between three replicates is very high to say with certainty that there is no difference in *GFP* activity. I would suggest repeating these experiments to increase statistical significance. But it is far more important to assess the RNA levels of these constructs. Given that an initiating ribosome complex at IRES may represent a roadblock to the scanning ribosomes it is also possible that initiation upstream may be stimulated by the PIC stacking as in ref 22 and in Dinesh-Kumar and Miller 1993 *Plant Cell* (10.1105/tpc.5.6.679). I am not sure how this could be ruled out, hence the authors at least should discuss this mechanism as an alternative to their chosen explanation.

The Reviewer brings up many good points in this commentary and we are grateful for the obvious care that has gone into reviewing our manuscript. Each major point is discussed in detail below.

- a. The experiments with *SINKFEL*-IRES-*GFP* are very provocative. However, the authors need to rule out the possibility that the effect that they observed is not due to changes in RNA levels. Poorly translated RNAs are often degraded more rapidly than highly translated. Thus, efficient translation from an active IRES may stabilize mRNA enabling detection of *SINKFEL* peptide even if its ORF translation efficiency does not change.

We agree with the Reviewer that it is important to assess the mRNA levels of different IRES constructs even though we used RNA transfection. We have followed the Reviewer's experimental suggestion and measure the half-life of IRES reporters in transfected cells. As shown in **Figure 4** of this letter, we observed comparable turnover of IRES reporters at the same time of flow cytometry measurement. We have now added this new result as Supplementary Fig. 2h in the revised manuscript.

Figure 4. HEK293 cells were transfected with indicated mRNA reporters followed by RT-qPCR at indicated time points. $n = 3$ biological replicates; Error bars indicate SEM.

- b. A suggestion that *SINKEL* uORF could be translated as a result of “forward scanning of the PIC looped from IRES” is an interesting one, but I do not understand how reversing polio IRES would rule it out. Probably the strongest support for the authors interpretation is that the increase of the spacer length between *SINKFEL* and *GFP* leads to *SINKFEL* reduction while *GFP* stays the same (right plots in fig 2e). However, I am not convinced that this is the case, looking at FACS plots, *GFP* is also reduced and for the plots on the right the variations for *GFP* activity between three replicates is very high to say with certainty that there is no difference in *GFP* activity. I would suggest repeating these experiments to increase statistical significance. But it is far more important to assess the RNA levels of these constructs. Given that an initiating ribosome

complex at IRES may represent a roadblock to the scanning ribosomes it is also possible that initiation upstream may be stimulated by the PIC stacking as in ref 22 and in Dinesh-Kumar and Miller 1993 *Plant Cell* (10.1105/tpc.5.6.679). I am not sure how this could be ruled out, hence the authors at least should discuss this mechanism as an alternative to their chosen explanation.

We apologize for the lack of clarity in describing the results from the sandwiched IRES construct. First of all, the construct only has a non-functional cap analog at the 5' end. Therefore, the initiation must start from the IRES between SIINFEKL and GFP. However, it has been suggested that the IRES-recruited ribosome could “jump” to the 5' end of mRNA (Paek *et al.* PNAS 2015, 112), permitting the translation of SIINFEKL in a forward scanning manner. To exclude this possibility, we reversed the IRES element to maintain the same RNA structure but with different orientation. The negative signals indicate that there is no PIC “jumping” or “looping” from the IRES.

To fully exclude the RNA “looping” possibility, we designed another IRES control by introducing a stem loop near the 5' end, which would block looping mechanism but not the 3'→5' scanning. Remarkably, presence of a stable secondary structure minimally affects the SIINFEKL translation (**Figure 5a** of this letter, left panel).

To thoroughly address the Reviewer's concern, we introduced an out-of-frame AUG after the first in-frame AUG of SIINFEKL. This design allows us to distinguish the direction of AUG selection between 5'→3' vs. 3'→5'. Remarkably, the 25D1 signal was abolished in the presence of the out-of-frame AUG (**Figure 5b** of this letter, right panel), suggesting that the translation of SIINFEKL relies on the downstream ribosome via backward sliding.

Collectively, these results strengthened the conclusion that ribosome is capable of bi-directional scanning. We thank the Reviewer's insightful comments that led us to delve deeper into the non-linear scanning mode. We have now presented these new results as Fig. 2f in the revised manuscript and described them accordingly in the main text.

4. *Read length analysis.* The authors consider slot-in and thread-in models as mutually exclusive, however, I don't see why we should exclude a possibility that both modes take place. This would explain the pattern observed in figure 3c without a need for backward scanning. Overall I find this analysis somewhat confusing and not convincing as asymmetric heterogeneity of footprint reads could occur due to the heterogeneity of the complexes, e.g. depending on the presence of a specific initiating factor a footprint may be shorter or longer either at the 5' or 3' side location. The authors interpretation of

Figure 5. Flow cytometry of HEK293-K^D cells transfected with mRNA reporters with the polio IRES sandwiched by SIINFEKL and GFP. **(a)** A stem loop with different stability was placed near the 5' end. **(b)** An out-of-frame AUG was inserted after the main AUG (bottom panel).

read lengths changes in eIF4A1 KO sounds too far reaching, there could be lots of different factors responsible for the observed changes.

We apologize for possible overstatement of the PIC-seq results shown in Fig. 3c. The Reviewer's idea about the co-existence of "slot-in" and "threading" models is attractive. However, "slot-in" and "threading" mechanisms rely on completely different actions of eIF4E, the cap-binding protein. The "slot-in" model is in line with the observed longer footprints at the 5' end (TSS). By contrast, "threading" is not expected to give rise to longer footprints at the 5' end. Finally, if the same mRNA could be translated via either "slot-in" or "threading", the AUG codon near the 5' end would be selected as efficiently as the one downstream.

Nevertheless, we agree with the Reviewer that the heterogeneity of ribosome complexes likely contributes to the size heterogeneity of footprints. In the absence of eIF4A, we were surprised to find longer reads in 5'UTR. This could be explained by the closed conformation of PIC that is expected to protect longer mRNA regions. To our view, the read length difference is quite significant when the mean read length is considered (from 28-nt to 35-nt). In particular, this difference can be seen from other independent datasets.

We understand that the footprint size heterogeneity by no means proves bi-directional scanning. Since other Reviewers raised the similar concern, we have now added a new paragraph in the main text to include all possible scenarios for PIC-seq data interpretation. We hope our work, once published, will trigger future investigations of scanning dynamics using techniques such as single molecule measurement.

5. *eIF4A1 KO. "Unexpectedly, messengers with ultra short 5'UTR were more sensitive to eIF4A1 depletion than those with long 5'UTR (Fig. 4b and Supplementary information, Fig. S4f)."*

I don't see that at all in Fig. 4b. The statistical significance changes, among 2, 3 and 5 nt leaders, but this is due to technical variation, not due to real changes. Why reporters with longer UTRs are not shown? Similarly for S4f, there are differences in activities of short UTRs for both KO and WT, but is this difference higher for shorter UTRs? Unclear if there is a difference between 2, 3 and 5 and longer ones are not shown.

We agree with the Reviewer that it is important to include a 5'UTR longer than 20-nt in Fig. 4b and Fig. S4f. To address the Reviewer's concern, we have repeated this experiment by including a mRNA reporter with β -globin 5'UTR. In cells lacking eIF4A, we observed reduced translation of Fluc with β -globin 5'UTR (**Figure 6** of this letter). This is consistent with the role of eIF4A in global mRNA translation, i.e., silencing eIF4A reduced protein synthesis as evidenced by polysome profiling and puromycin labeling (Figure S4 of the manuscript). The relative

Figure 6. Translation efficiency of Fluc mRNA reporters with ultra-short 5'UTR in MEF cells with or without eIF4A1 knock-down. Fluc activities in the absence of eIF4A1 were normalized to the control. Error bars: mean \pm SEM; $n = 4$ biological replicates. One-way ANOVA, * $p < 0.05$; ** $p < 0.01$.

resistance of the reporter with 20-nt 5'UTR to eIF4A knockdown could be explained by the “slot-in” model that directly locates the P-site of 40S near the start codon. With minimal scanning, translation of the mRNA reporter with 20-nt 5'UTR is the least eIF4A-dependent. We have updated Fig. 4b in the revised manuscript.

The Reviewer is quite correct that, in cells with or without eIF4A1 knockdown, the difference between short 5'UTR (2, 3, 5 nt) and long 5'UTR (20 nt) is rather small. To address the possibility of technical variation, we silenced eIF4A2, which reduced the translation of all mRNA reporters in a uniform manner, regardless of the 5'UTR length (**Figure 7A and 7B** in this letter). Therefore, the differential effect of eIF4A1 toward short and long 5'UTR is quite specific. In the revised manuscript, we have presented these new results as Supplementary Fig. 4h.

Figure 7. (A) Immunoblotting of MEF cells with or without eIF4A2 knockdown **(B)** Translational efficiency of Fluc mRNA reporters with different 5'UTR in MEF cells with or without eIF4A2 knockdown. The right panel shows the ratio of Fluc levels before and after eIF4A2 knockdown. Error bars; mean \pm SEM; n=3 biological replicates. One-way ANOVA, ns, not significant.

6. Oscillations and the suggestion that backwards scanning increases translation at upstream starts.

“Since PIC oscillation occurs during the entire course of scanning, the back-and-forth excursions is expected to increase the dwell time of PIC in 5'UTR and facilitate the recognition of upstream start sites (uTIS).”

I don't see how. Whether PIC moves unidirectionally (only forward) or bidirectionally (say two steps forward, one backward) the relative time that ribosome spend at downstream sites and upstream sites would be the same. Hence, such oscillations do not increase the chances of uTIS recognition. Nonetheless this “Remarkably, PatA treatment for 2h resulted in a global increase of 5'UTR read density relative to the CDS (Fig. 5b, right panel).”

is a poor support for the above anyhow as increase of ribosome density in 5'UTRs is a general phenomenon observed in many stresses and could be explained in a number of ways. Increased amount of free ribosomes, reduced stringency of start selection due to subsequently limited number of available initiation factors are only some of many plausible explanations. Contrary to the authors descriptions the effect is seen for both PatA and Hippusinol, it is just more pronounced for PatA at earlier time points (Fig. 5b), hence I wonder if it relates more to the severity of the stress rather than the specificity of the compound.

We appreciated the Reviewer's effort to improve our manuscript. The Reviewer might be confused by our interpretation of the results shown in Fig. 5B, possibly due to the lack of clarity on our side. A growing body of evidence suggests that the scanning speed influences the start codon selection. For instance, slowing down the ribosome scanning by introducing a secondary structure in 5'UTR would increase the selection of upstream non-canonical start codons that otherwise would be skipped (Medenbach *et al.* Cell 2011, 145). This is primarily due to the increased dwell time of PIC in 5'UTR. For the same reason, bidirectional scanning would increase the chance of selecting upstream start codons. Importantly, this would readily change the opportunity between upstream and downstream start site selection because, once an upstream site is selected, the downstream site is masked by elongating ribosomes.

With respect to the alternative explanation of 5'UTR ribosome density, we agree with the Reviewer that many stress conditions could lead to non-canonical translation. But to our knowledge, stress-induced alternative translation is limited to a few mRNAs. In our case, the clear difference between PatA and Hippu is global and thus cannot be explained by stress conditions. It is worth mentioning that Hippu is more toxic than PatA, but it barely increases the 5'UTR read density. More recently, we have uncovered numerous upstream start codons in a global manner by using low doses of PatA (unpublished results). We believe (and hope the Reviewer will concur) that PatA differs from Hippu by acting on eIF4A1 in a distinct manner, resulting in increased 5'UTR translation. In the revised manuscript, we have rephrased several sentences in the main text to ensure clarification.

Reviewer #3:

In the manuscript "Bidirectional ribosome scanning controls the stringency of start codon selection" the authors aimed to understand how ribosomes are recruited to mRNA and whether the process of PIC scanning is unidirectional or bi-directional. The study argues that ribosome are recruited by the slot-in mechanism and then that the PIC undergoes oscillations (backsliding) aided by the ATPase and helicase activities of eIF4A. The study is thorough and addresses an important unresolved problem in translation initiation. However, while we agree with the authors that the evidence for some backsliding is fairly convincing and that recruitment through slot-in is very likely to occur, we are not altogether convinced that the "threading" model can be entirely discarded. As also pointed out in the discussion section, the two models are not mutually exclusive. We also believe the manuscript would benefit from more thoroughly addressing previous results and evidence that has been presented in favor of this model and discuss how these should be interpreted under this new model.

We thank the Reviewer's time and effort in reviewing our manuscript. We appreciate the Reviewer's agreement that our work addresses a fundamental biological question. We agree that a direct evidence to exclude the "threading" model is still lacking. Since the Reviewer #2 raised the similar concern, we have revised our manuscript by including the co-existence possibility of "slot-in" and "threading" during translation initiation of different types of mRNAs.

We are also grateful to the Reviewer for reminding us of other relevant literatures in supporting both "slot-in" and "threading" models. We are glad to have this opportunity to further improve our manuscript.

Major issues:

1. *TISU elements are only very briefly mentioned in the discussion stating that these are "unlikely functional in the form of a linear thread". What is the evidence for this statement? The existence of TISUs is a significant argument in favor of threading and should therefore not be casually dismissed but addressed through experiments to understand how the proposed backsliding model can explain the high efficiency of these cap-proximal initiators. Would you e.g. see initiation downstream of TISU elements as would be predicted by slot-in and backsliding? Would eIF4A1 KD have any effect? In general, the surrounding context of initiation codons seems to be somewhat ignored in the manuscript.*

We apologize for the incorrect statement regarding the TISU-mediated translation initiation. TISU is a 12 nt element directing scanning-independent translation. With the core sequence SAASATGGCGGC (S = C or G), TISU appears to interact with ribosomal proteins (such as RPS3 and RPS10e), thereby securing the 48S PIC at the correct position (*Mol Cell Biol* 2017, 37). As pointed out by the Reviewer, this can be achieved by threading from the 5' end. However, "threading" itself can be viewed as the scanning process, albeit the driving force remains elusive. The presence of TISU likely slows down "threading" once it encounters RPS3, resulting in so-called scanning arrest. Notably, this can be equally accomplished by "backsliding" after slot-in. In fact, the size

of TISU is nearly perfect for the ribosome A-site to directly land on the start codon after cap-dependent slot-in. Based on the footprint size at 5' end of mRNA (or TSS), we speculate that the presence of TISU stabilizes the 48S around the start codon with minimal scanning.

In our study, we used 5'UTR either much shorter (such as 5 nt) or longer than TISU (such as 20 nt) in order to test the scanning process after loading. Therefore, they are fundamentally different from TISU. Since TISU involves minimal scanning, we believe (and hope the Reviewer will concur) that experiments using TISU unlikely offer mechanistic insights into the scanning process.

In terms of the surrounding context of start codons, we primarily used optimal Kozak sequence context to avoid variation during start codon recognition. This would allow us to test the sole contribution of 5'UTR length to start codon selection when the PIC landed either downstream or upstream of the start codon.

We thank the Reviewer for reminding us of TISU-mediated translation initiation. With an average length of 12 nt, TISU likely represents a unique 5'UTR that requires minimal scanning for start codon selection. In the revised manuscript, we have extensively discussed the TISU elements in the main text.

2. Similarly, there is no mention of the studies of eIF1 which is the factor thought to inhibit translation initiation proximal to the cap under the “threading” model. A discussion of how to reconcile these prior studies would be shed light on how to interpret the presented results.

We appreciate the Reviewer’s insightful comments regarding the potential role of eIF1 in translation initiation. It is interesting to find that eIF1-eIF4G competes with eIF4E-eIF4G in promoting TISU translation (*Mol Cell Biol* 2018, 38), suggesting the “slot-in” mode for TISU translation. In fact, the original paper also follows “slot-in” instead of “threading” for start codon selection (see Fig. 6). As we mentioned above, the presence of TISU permits start codon recognition with minimal scanning. It is possible that eIF1 antagonizes eIF4E to destabilize the 48S position, thereby causing leaky scanning.

To fully address the Reviewer’s concern about eIF1 that is highly relevant to our work, we knocked down eIF1 from HEK293 cells (**Figure 8a** of this letter). Notably, silencing eIF1 had marginal effects on global protein synthesis and cell growth. We then examined the translation of mRNAs with varied 5'UTR length and observed a similar reduction in mRNA translation, regardless of the 5'UTR length (**Figure 8b** of this letter). This result is consistent with the previous findings that depletion of eIF1 suppressed both TISU- and scanning-dependent translation.

Figure 8. (a) Immunoblotting of HEK293 cells with or without eIF1 knockdown. **(b)** Translation efficiency of Fluc mRNA reporters bearing varied length of 5'UTR in cells with or without eIF1 knockdown. Error bars: mean \pm SEM; n = 3 biological replicates.

In considering all those prior studies, we are now more confident about the hypothesis that ribosome is capable of bi-directional scanning. Thanks for the Reviewer's insightful comments, we have now expanded the discussion section in the revised manuscript to include all possible scenarios.

3. Fig 2b, shows a two-fold decrease in fluc activity on eIF4E crosslinking, which implies that translation is substantially inhibited by the tethering of the cap-protein (eIF4E). This "lost translation" could very well be occurring via the threading mechanism and it is therefore a too strong statement to claim that "This result cannot be explained by the "threading" model ...". It is only correct that it cannot be explained by "threading" being the ONLY mode of initiation.

The Reviewer might be confused by our interpretation of the eIF4E crosslinking results shown in Fig. 2b, partly due to the lack of clarity in our original description. The difference between eIF4E crosslinking and non-crosslinking could be due to multiple reasons. First, m⁷S⁶G is a suboptimal cap analog compared to the normal m⁷G cap, as shown in Supplementary Fig. 2b. Second, UV exposure at 360 nm is known to cause non-specific effects, which likely contribute the decrease of Fluc activity. Third, crosslinking inevitably limits eIF4E recycling that might cause general reduction of translation. Therefore, the observed translation reduction under eIF4E crosslinking is quite expected, but should be independent of mRNA reporters. Since mRNA reporters with ultra-short 5'UTR showed much higher sensitivity to eIF4E crosslinking as shown in Fig. 2c, it is likely that ultra-short 5'UTR relies on backsliding for start codon selection.

In terms of the co-existence of "threading" and "slot-in", we agree that it is possible for different types of mRNAs but not the same mRNA. If the same mRNA can be translated by either mechanism, it cannot explain the "lost translation" under eIF4E crosslinking. The reasoning is quite straightforward: when one choice (such as "threading") is not possible, the second choice (such as "slot-in") is expected to be fully compensatory. It is hard to believe that the same mRNA would form two distinct pools undergoing different modes of translation. We thank the Reviewer for understanding.

4. The same argument can be raised for Fig 2C where an alternate interpretation can be that the decrease in fluc activity for mRNAs with ultra-short 5'UTR upon eIF4E cross-linking is due to the mRNA not being able to enter the ribosomal mRNA channel through "threading".

The Reviewer brought up an interesting point in this commentary, which is relevant to the issue mentioned above. It is important to note that the results in Fig. 2b and 2c need to be considered in parallel. Based on the Reviewer's reasoning, if the "lost translation" for ultra-short 5'UTR under eIF4E crosslinking is solely due to blocked "threading", then an equal inhibition is expected to occur on β -globin 5'UTR shown in Fig. 2b. Apparently this is not the case because long 5'UTR is much less sensitive to eIF4E crosslinking. We would like to emphasize that neither "threading" nor "slot-in" model alone could explain all the results, which prompted us to propose a bi-directional scanning model. We believe (and hope the Reviewer will concur) that the non-linear scanning model is supported by multiple lines of independent evidence.

In the revised manuscript, we have made further clarifications in the result interpretation of Fig. 2b and 2c.

5. It is unclear why the “bump” in 3’ ends is expected by slot-in followed bi-directional scanning. Is bidirectionally only occurring right after slot-in? If bi-directional scanning occurs at the same probability throughout the 5’UTR the 3’ distribution should not necessarily differ close to the cap, it should rather increase up to the site of slot-in and from there be relatively uniform throughout the 5’UTR. It is also not immediately obvious that other models could not produce this pattern. A similar bump could result without back-sliding if slot-in is not precise down to a single base, but the 5’ end remains protected, or under the threading model if the speed is not uniform and/or if there is a detectable loss of scanning complexes across the UTR.

We agree with the Reviewer that the distribution pattern of PIC footprints near the 5’ end could be explained by multiple mechanisms. However, we’d like to argue the Reviewer’s alternative explanation one by one.

First, we agree with the Reviewer that, if bi-directional scanning occurs throughout the 5’UTR, a uniform pattern is expected from the 5’ end. However, the initial back sliding after “slot-in” is distinct from downstream back scanning. The presence of eIF4E at the 5’ end represents a constraint for back sliding, likely contributing to the 3’ end “bump”. The 5’ end constraint does not exist for downstream region of 5’UTR.

Second, we like the idea of non-precise “slot-in” that could form a “bump” instead of a “peak”. However, the presence of small footprints at the 5’ end supports bi-directional scanning after “slot-in”.

Third, the notion of non-uniform speed of “threading” is interesting and it is expected to cause a similar “bump”. However, under the “threading” scenario, the rate-limiting step is in the beginning of 5’UTR when the 5’ end is still buried inside the mRNA tunnel. If so, a “bump” is expected to form near the TSS. The presence of bump at the downstream 30 nt strongly suggests that the 5’ end cap already emerges from the PIC.

Nevertheless, the Reviewer’s comments are extremely insightful. In the revised manuscript, we have considered all possibilities during the description of the results shown in Fig. 3a. As a result, a new paragraph is now added into the main text.

6. While the manuscript makes a point of the longer reads at the TSS it should also be noted that both TCP-seq and RCP-seq revealed a significant number of shorter fragments (which could probably be even more abundant if not size-selected and lost due to mapping issues with very short reads). While this is consistent with both backsliding and threading, it would under the backsliding model suggest that a substantial(?) number of ribosomes slide entirely off the mRNA. At the very least this possibility should be addressed and discussed.

We thank the Reviewer’s effort to improve our manuscript. The Reviewer is correct that both TCP-seq and RCP-seq revealed many short reads. We agree with the Reviewer that those short fragments could be explained by either “threading” or “back-scanning” model. In the revised manuscript, we have incorporated both scenarios in describing

the pattern of PIC footprints near the 5' end. As mentioned above, a new paragraph is now added into the main text.

Minor issues:

Page 3, line 72: scanning PIC encounters “servers” should be serves

Page 3, line 82: negatively affects should be negatively affecting

Page 10, line 274: Fig.5a, left panel should be Fig.5a, right panel

Page 11, line 303: celled should be called

Supplementary fig 4h and 4i are not referred to in the main text.

We appreciate the Reviewer’s obvious care in reviewing our manuscript. We are glad to have this opportunity to correct these errors in the revised manuscript.

Closing words

We thank all the Reviewers for providing constructive comments that have helped improve our manuscript. We have thoroughly revised our manuscript and addressed all of the concerns. In particular, we have performed additional experiments to strengthen our original conclusions. We believe the revised manuscript has been much improved. We wish to convey our most sincere thanks to all the Reviewers and the Editor for considering our work for publication in ***Nature Communications***.

Reviewers' Comments:

Reviewer #1:

Remarks to the Author:

The authors have presented a detailed rebuttal, presented new experimental evidence and revised parts of the manuscript text to address the reviewers' concerns. I believe that this study is now ready for publication.

Reviewer #2:

Remarks to the Author:

The authors addressed my comments in full and significantly improved the manuscript.

I still remain sceptical regarding the authors' interpretation of PatA and Hippu data as well as of the suggestion for oscillations.

The authors argued that the differences between PatA and Hippu are global while stress-induced alternative translation is limited to few mRNAs. This is factually wrong, the stress-induced alterations in translation are global, specific RNAs escape these changes, but the translation of the most RNAs is altered. Hence, I still remain of the opinion that the reason for the observed difference could be quantitative rather than qualitative.

As for the oscillations, I understand how slowing scanning ribosomes could increase initiation at non-AUG codons, there is indeed plenty of evidence supporting this. What I do not understand is how two steps forward and one step backwards (at a time) is slower than simply one step forward (at a time).

These disagreements primarily relate to data interpretation. I do not think that they warrant another round of revision. I just would suggest that the authors should think carefully about how to describe these points in their final text to avoid overinterpretation.

Reviewer #3:

Remarks to the Author:

The authors have addressed most of our points, but the TISUs concerns are still somewhat casually dismissed. We believe it is important to distinguish between two different claims: 1) does back-sliding occur? and 2) does threading `_never_` occur? We perceive (1) as the main message of the article.

While we agree with the reply to Q1 that "experiments using TISU unlikely offer mechanistic insights into the scanning process." this was not our point. We believe it could offer insight into backsliding+slotting vs threading question since this is a specific instance where threading is a suggested pathway. For instance

1) Can TISU transcripts still initiate translation on cross-linked caps as predicted by slot-in/backsliding (but impossible under threading)?

2) Does the TCP/RCP read density "bump", suggesting inaccurate slot-in followed by backsliding, occur on TISU transcript or is the read distribution very sharp as predicted by the optimal distancing mentioned in the main text: "With a 12 nt core sequence, TISU is nearly perfect for the ribosome A-site to land near the start codon"^[1]. Does shorter 5' fragments from RCP/TCP seq occur on TISU transcripts or mostly long fragments perfectly over the ATG as predicted by slot-in?

3) In fact, it would be interesting to see how TCP/RCP distributions behave on cap-cross-linked

libraries. Do all short fragments at the 5' end completely disappear?

We do not think these points are necessary to address "back-sliding" however, and therefore not crucial to the manuscript.

Reviewer #1:

The authors have presented a detailed rebuttal, presented new experimental evidence and revised parts of the manuscript text to address the reviewers' concerns. I believe that this study is now ready for publication.

We thank the Reviewer's support for publishing our manuscript.

Reviewer #2:

The authors addressed my comments in full and significantly improved the manuscript.

I still remain sceptical regarding the authors' interpretation of PatA and Hippu data as well as of the suggestion for oscillations.

The authors argued that the differences between PatA and Hippu are global while stress-induced alternative translation is limited to few mRNAs. This is factually wrong, the stress-induced alterations in translation are global, specific RNAs escape these changes, but the translation of the most RNAs is altered. Hence, I still remain of the opinion that the reason for the observed difference could be quantitative rather than qualitative.

As for the oscillations, I understand how slowing scanning ribosomes could increase initiation at non-AUG codons, there is indeed plenty of evidence supporting this. What I do not understand is how two steps forward and one step backwards (at a time) is slower than simply one step forward (at a time).

The Reviewer's remaining concern lies in the argument we made in our previous rebuttal, in which we stated that the differential effects between PatA and Hippu were global. It is well-established that the stress-induced translational alternations include global shutdown as well as selective translation of specific mRNAs. It is not our intention to claim that the latter is more dominant than the former. We were merely stating that the increased 5'UTR reads in the presence of PatA cannot be simply explained by the stress response, because Hippu treatment did not increase the 5'UTR read density at all. Nevertheless, we agree with the Reviewer that the observed difference is likely quantitative rather than qualitative.

Regarding the oscillations, the Reviewer raised an interesting kinetics issue. If the movement speed is the same, then two step forward and one step backward is definitely slower than one step forward. Like the chess game, two step forward and one step backward only moves the pawn one step. By contrast, another pawn can move three steps if no backward. We thank the Reviewer for understanding.

Reviewer #3:

The authors have addressed most of our points, but the TISUs concerns are still somewhat casually dismissed. We believe it is important to distinguish between two different claims: 1) does back-sliding occur? and 2) does threading _never_ occur? We perceive (1) as the main message of the article.

While we agree with the reply to Q1 that “experiments using TISU unlikely offer mechanistic insights into the scanning process.” this was not our point. We believe it could offer insight into backsliding+slotting vs threading question since this is a specific instance where threading is a suggested pathway. For instance

1) Can TISU transcripts still initiate translation on cross-linked caps as predicted by slot-in/backsliding (but impossible under threading)?

2) Does the TCP/RCP read density “bump”, suggesting inaccurate slot-in followed by backsliding, occur on TISU transcript or is the read distribution very sharp as predicted by the optimal distancing mentioned in the main text: “With a 12 nt core sequence, TISU is nearly perfect for the ribosome A-site to land near the start codon”. Does shorter 5' fragments from RCP/TCP seq occur on TISU transcripts or mostly long fragments perfectly over the ATG as predicted by slot-in?

3) In fact, it would be interesting to see how TCP/RCP distributions behave on cap-cross-linked libraries. Do all short fragments at the 5' end completely disappear?

We do not think these points are necessary to address "back-sliding" however, and therefore not crucial to the manuscript.

We appreciate the Reviewer's effort to improve our manuscript. We did not conduct cap cross-linking for TISU transcripts because most TISU elements are longer than 12 nt. We reasoned that the result will be less informative because even non-TISU mRNAs are translatable as long as the 5'UTR is longer than 12 nt.

With regard to the TCP/RCP read density for TISU transcripts, we agree that it will be interesting to know the “bump” position. However, the limited number of TISU transcripts do not have sufficient reads for metagene analysis. Nevertheless, we selected transcripts with short 5'UTR (< 30 nt) followed by the similar analysis as long 5'UTR. Interestingly, there is a sharp peak corresponding to the initiating ribosome with the P-site at AUG (**Figure 1** of this letter). Since not all those short 5'UTRs belong to TISU, we did not include this result in the revised manuscript to avoid confusion.

It is noteworthy that cap cross-linking can only be done in vitro. Therefore, it is not possible to conduct TCP/RCP analysis on cap cross-linked libraries. Regardless, we completely agree with the Reviewer that follow-up studies are warranted to investigate the unique feature of TISU mRNAs.

Figure 1. (A) Reanalysis of TCP-seq data sets obtained from *S. cerevisiae* (Archer et al). The left panel shows the aggregation plots of 5' end (green) and 3' end (red) of reads on mRNAs with 5'UTR >30 nt aligned to TSS. The right panel shows the aggregation plots on mRNAs with 5'UTR <30 nt aligned to TSS. **(B)** Reanalysis of RCP-seq data sets obtained from zebrafish (Giess et al). The second “bump” is highlighted in red shades.